



# Spatial identification of regions at risk to multi-hazards at pan European level: an implemented methodological approach

Tiberiu-Eugen Antofie, Stefano Luoni[1], Alois Tilloy,[1] Andrea Sibilia[1], Sandro Salari[1], Gustav Eklund[1], Davide Rodomonti[1], Christos Bountzouklis[1] and Christina Corbane[1]

1 European Commission, Joint Research Centre,  Ispra, 21027, Italy

*Correspondence to*: Tiberiu Antofie Eugen (tiberiuantofie@yahoo.com)

**Abstract.** The Disaster Risk Management Knowledge Centre (DRMKC) is hosting a web platform – the Risk Data Hub -

intended to improve the access and sharing of curated EU-wide risk data, tools and methodologies for fostering Disaster Risk Management (DRM) related actions. Within the DRMKC's Risk Data Hub (RDH) development, we integrate a methodology for the identification of regions with multi-hazard potential impact at pan European level. With this study we present the methodological approach and we stage it as one fundamental development in support of DRM decision-making at national and subnational level.

We adopt a meta-analysis approach, combining independent tests (the single hazards' exposure hotspots), which seeks to solve the problem of "insignificant results" and provides an objective "statistical proof" of the multi-hazard potential of a region.  We support these results through a validation process which considers empirical data as explanatory variables.

Presenting an implemented methodology, scalable down to local subnational level, that reveals types of assets at risk to multiple hazards and their location, we take one further step towards the identification of the disaster risk management

pathways in multi-hazard assessment.

The outcome of this study will provides valuable input and will assist national authorities on the integration of multi-hazard analysis in their National Risk Assessments and Disaster Risk Management plans.




## 1. Introduction

The results of a ''Needs and Gaps'' analysis performed as part of the preparation of the European Commission Staff Working Document – ''Overview of Natural and Man-made Disaster Risks the European Union may face'' (2014[1] , 2017[2] ,2020[3]), concluded that a gap in knowledge and data availability exists for multi-hazard assessments. A number of international
frameworks such as Hyogo Framework for Action (UN-ISDR, 2005) or Sendai Framework for Disaster Risk Reduction 2015–2030, have endorsed the multi-hazard approach for Disaster Risk Reduction.

It is now well recognized in the research community that for an adequate understanding of disaster risk potential within a region it is essential to move from single hazard to multi-hazard approach (Marzocchi et al., 2009, Kappes et al., 2012, Gill and Malamud., 2014, Tilloy et al. 2019, Ward, P. J., et al 2022). The hazard interrelations can lead to a combined impact that
is different from the sum of each hazard's impacts separately. In order to assess the potential hazards and the risk to which a region is exposed, some studies combined independent analysis of single hazards (Granger et al., 1999; van Westen et al., 2002; Greiving et al., 2006; Grünthal et al., 2006; Marzocchi et al., 2012, Forzieri et al., 2016) and superposed natural hazards over a region (multi-layer hazards). Others studies considered hazard interactions (Tarvainen et al., 2006; Han et al., 2007; De Pippo et al., 2008; Kappes et al., 2010; van Westen et al., 2014, Liu et al., 2017; Sadegh et al., 2018; Gill et al.,
2020). Often, these assessments are based on case studies within limited spatial extension, addressing a limited number of perils/hazards and addressing specific sectors (Ciurean et al., 2018, Tilloy et al., 2019).

One development that addresses these challenges, is the DRMKC Risk Data Hub platform (DRMKC RDH) of the Disaster Risk Management Knowledge Centre (DRMKC). The platform improves the access and sharing of curated European-wide risk data and methodologies being a fundamental tool in support of the DRM[4] and CCA[5] actions at national and subnational
level. Within the DRMKC Risk Data Hub development, we propose a methodology which is accessible, scalable and replicable even at subnational and local level for the identification of regions at risk to multi-hazards.

The multi-hazard methodological approach becomes the main goal of this study, focused on four major challenges: (1) identification of regions with significant multi-hazard potential; (2) establishment of the impact relation assets – multiple hazards, (3) quantification of multi-hazard potential impact and (4) transferability of the method. These challenges are
further constrained by the wide scale of our analysis (European coverage), the alignment to a common hazard definition and their practical implementation on the online web platform, the DRMKC RDH.

We argue, in this study, for a novel methodology (1) that identifies, at pan-European scale, the regions (Local Administrative Units - LAUs) at risk to multi-hazards with high level of significance. We adopt a meta-analysis approach, combining independent tests (the single hazards' exposure hotspots), which seeks to solve the problem of "insignificant results" and

[1] EUR-Lex - 52014SC0134 - EN - EUR-Lex (europa.eu)
[2] https://ec.europa.eu/echo/sites/echo-site/files/swd_2017_176_overview_of_risks_2.pdf
[3]https://ec.europa.eu/echo/sites/echo-site/files/overview_of_natural_and_man-made_disaster_risks_the_european_union_may_face.pdf
[4] https://civil-protection-knowledge-network.europa.eu/knowledge-network-science/data-tools
[5] eur-lex.europa.eu/legal-content/EN/TXT/PDF/?uri=CELEX:52021DC0082 (pg. 6)



provides an objective "statistical proof" of the multi-hazard potential of a region. This is the first study that uses spatial patterns (clusters/hotspots) and meta-analysis for this purpose.

Furthermore (2), we show that the proposed methodology allows for the detection of the regions at risk to multi-hazards, differently, as function of the typology of the assets. This is important as it directly reveals relationships between assets types and threats, valuable for the identification of the disaster risk management pathways in multi-hazard assessment (Ward, P.

J., et al 2022). This is the central aspect of the multi-hazard analysis presented in this study, which considers the relation of single asset (population and the residential built-up respectively) to the multiple hazards: landslide, coastal flood, river flood, earthquake, wildfires and subsidence.

The quantification of multi-hazard potential impact (3) is based, for this study, on the total assets found in the regions at risk to multi-hazards with high level of significance. This areal dimension approach (Hewitt and Burton, 1971), omits a detailed

level of study that could more accurately examine the spatial coincidence, trigger relations or cascading effects when quantifying the impacts from multi-hazards. Nevertheless, we argue that our methodology succeeds in describing the "hazardousness" level of a region offers a generalized spatial understanding of where the specific assets are exposed to multiple hazards and what hazard becomes accountable for a potential impact.

We also show that in contrast to other studies, the transferability of the developed methodology (4) is not limited due to the

reliance on case-study-specific data and methods. The methodological approach described in this study it is already implemented on the DRMKC Risk Data Hub platform and uses the existing pan-European data hosted and shared through the platform.

We structure the study as follows: after the *Introduction* we describe the *Data and the methodology* used.(i) We describe the underlying exposure data and methodology that creates the bases for our single and multiple—hazard analysis, (ii) We

present the methodological approach used to find the hotspots for single hazard's exposure (iii) We present the meta-analysis methodology used to combine the hotspots of single hazards' exposure and to identify regions at risk to multi-hazards identified with high significance. In *Results* and based on these identified regions we provide a statistical analysis looking at different socioeconomic features: (iv) levels of economic development (high, middle high, middle low and low income regions - LAUs), (v) urbanization levels: rural and urban (according to URAU audit 2018 definitions across European

LAUs), (vi) We identify metropolitan areas at risk to multi-hazards (vii) and comparing city centres (city cores – C ) and Functional urban areas (FUA) levels in the metropolitan areas. Furthermore, a *Validation* exercise is performed followed by *Discussions* and *Conclusions*.






## 2. Data and methodologies

The methodological approach is presented in 3 steps: (1) we describe the underlying exposure data and methodology that creates the bases for our single and multiple—hazard analysis, (2) we present the methodological approach used to find the hotspots for single hazard's exposure,(3) we present the metadata analysis methodology used to combine the hotspots of single hazards' exposure and to identify regions with significant multi-hazard potential. A representation of the entire methodological chain is provided in fig. 1.

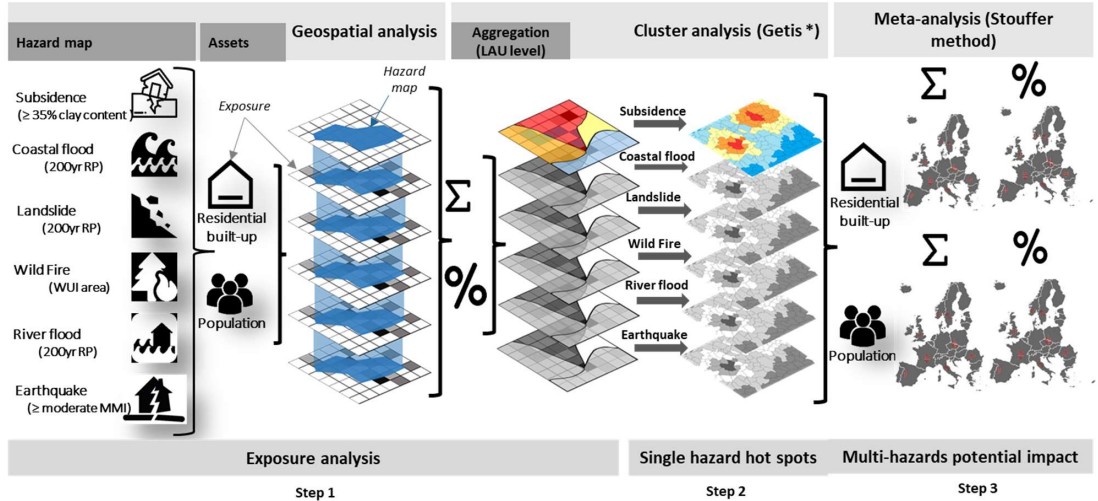

**Figure 1 Different steps of the methodological approach developed in this study**

### 2.1 The exposure data and methodology

### 2.1.1 The areal dimension

For this study the multi-hazard spatial coincidence is assessed at the level of areal dimension, represented by the Local Administrative Units (LAUs).

The LAUs are the finest hierarchical classification of statistical regions that together subdivide the European economic territory into regions. This dataset comes from the statistical office of the European Union (Eurostat) and represents the administrative units of municipalities and communes of Europe, version 2013. In the present study, the LAUs cover the European Union 28 and the European Free Trade Association (EFTA) countries.

These administrative entities are used as statistical areas for multi-hazard exposure and hotspot analysis as an approach meant to support disaster risk management activities. Administrative directives, organisations and operational services are


coordinated at the level of administrative entities and they become of high relevance when linked down to local level, challenging the gap in the scale of policy and scale of practice (Gaillard et al., 2013).

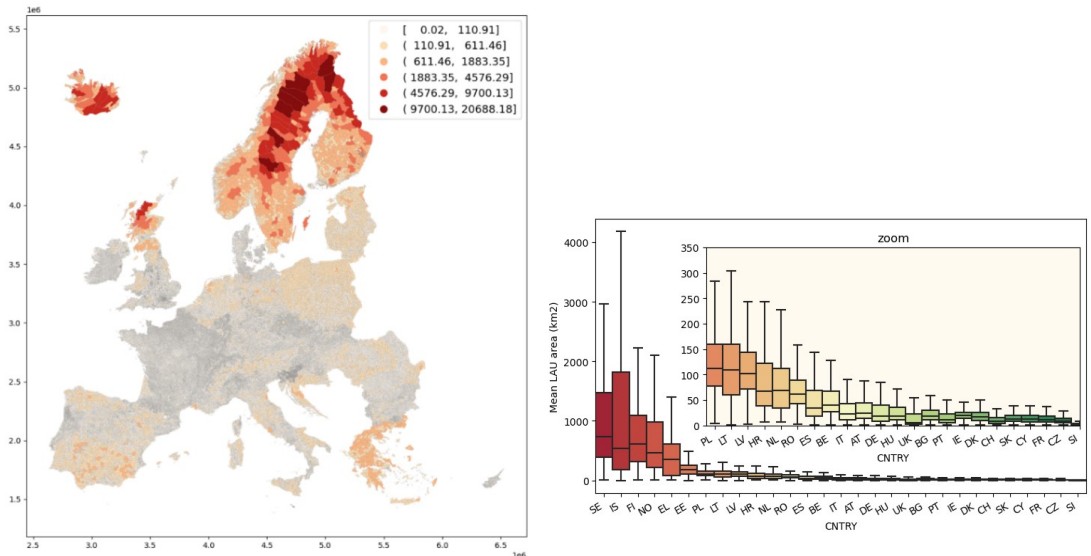

115

**Figure 2. Local Administrative Units area (km2), spatial distribution (left) and mean LAUs area per country (right)**

There are 122 034 LAUs considered as geographical statistical units on which the aggregations and statistical analysis are performed in this study. Their average area is 39.6 km$^2$, the maximum area is 20 688 km$^2$ (Kiruna, SWE) and the minimum is 0.2 km$^2$ (Thorpe Hamlet, UK). LAUs present heterogeneities across Europe in terms of area covered especially in northern

120 part of Europe (e.g. Scandinavia), even if they are rather homogeneously distributed within the national boundaries (fig. 2). Despite being a well-established geographic concept, the process of aggregating higher resolution data to larger administrative units comes with a potential source of error known as modifiable areal unit problem (MAUP). The two related issues to the MAUP, largely presented in the literature (Fotheringham and Wong, 1991; Jelinski and Wu, 1996; Openshaw, 1984) are the scaling and the zonation effect (Charlton, 2009). These are generally altering the variance structure of the data

125 when aggregated due to disconnection across scales and to different ways of subdividing the geographical space at the same scale (Stillwell et al., 2014). In order to minimize the MAUP effect, recommended practices (Su, 2011; Kwan, 2012) which are consistent with our approach focus on using smaller areal unit (e.g., LAUs rather than provinces or countries) for data aggregation. It reduces the potential errors of spatial pattern distortion without completely removing it.



### 2.1.2. Exposure data

The exposure data is built on the relationship hazard *(i)* - assets *(ii)*. We overlay spatial information about residential built-up and population with data describing hazard areas in order to define the assets exposure to single hazards. We than aggregate the exposure at the level of LAUs. We search for the significant hotpots of assets at risk from single hazards using two types of exposure aggregation:

- based on absolute values - the sum of the exposed asset
- based on relative values - as ratios or share of the exposure from the total amount of asset in a LAUs.

For the exposure to earthquake, due to the continuous spatial extent of the hazard area, we depict the relative aggregation schema using the density (or share of the exposure compared to the total area of the LAUs). The relative aggregation schema intends to address risk management strategies based on the cost efficient-measure while the absolute schema supports the risk management strategies that prioritize the most affected areas and people.

*(i)    Hazard Layers*

The hazard layers considered in this study represent areal extension rather than intensity. We do not use a probabilistic assessment but rather a deterministic approach selecting hazards with average temporal (frequency of occurrence) and spatial probability (susceptibility). A review of the hazard datasets and their characteristics is presented in Table 1. The motivations for their selection along with their usage in disaster risk assessments are presented in the sections dedicated to individual hazards in Supplementary material (Section 1 - Hazard data).

**Table 1. Description of the Hazards scenarios and datasets considered and their characteristics**

| Component | Scenario | Description | Spatial resolution | Data source |
|---|---|---|---|---|
| River flood | 1 event in 200yr RP | Areal extent of the river flood prone areas | 100m | EFAS (European Flood Awareness System), KULTURisk project |
| Landslide | High and very high susceptibility classes | Physical characteristics of various terrain factors that provides high predisposition to landslide occurrence (ELSUS 100 layer) | 200m | ESDAC |
| Coastal inundation | 1 event in 200yr RP | Areal extent of coastal inundation as extreme total water level (TWL) result of the contributions from the mean sea level (MSL), the tide and the combined effect of waves and storm surge. | 100m | HELIX project, JRC CoastalRiskandGAP-PESETAII projects |




| Earthquake | PGA >= 0.18 (g) for a probability of exceedance of 10% in 50 years (475yr RP) | Areal extent of PGA >= 0.18 (g) , equivalent of 'Moderate', 'Moderate to heavy' 'Heavy'', ''Very heavy' potential damage level of USG Intensity Scale | 1000m | SHARE project |
|---|---|---|---|---|
| Subsidence (from drought) | Soils with clay content greater than 35%. | Areal Extent of fine and very fine soil texture (particle < 2 mm size) and with clay content greater than 35%. | 1000m | ESDAC, IPL project |
| Forest fire | Wildland–Urban Interface area (WUI) | WUI areas within 10 km limit range from the historical  burned areas (2000-2016) | 100m | EFFIS based |

150

*(ii)     Assets layers*

As assets layer, we use the residential built-up from the European Settlement Map (ESM) (Florczyk, et. al., 2015) and residential population form the Global Human Settlement Layer (GHSL) (Freire, et al.2015). These are two main groups of assets that are present currently across all types of analysis within the DRMKC Risk Data Hub. The residential built-up is represented as built-up area (km2) and the population is amount of people within 100m x 100m grid cells.

In order to discriminate the residential typology for both built-up and population, the Corine Land Cover (CLC 2018) code 1.111 and 2.112 is used as the artificial explanatory layer.

### 2.2    Single hazard hotspots analysis

The study uses a hotspot analysis in order to identify clusters (concentrations) of regions - LAUs,  with assets at risk to single-hazard. The chosen approach enables the recognition of spatial patterns and trends which are not immediately apparent in raw data and which exhibit underlying spatial processes at work that are not the result of random processes (Getis and Ord., 1992). We argue that these spatial patterns (hotspots) once combined across multiple-hazards will describe the statistically significant multi-hazard potential of regions.

Various methods for combining single hazard data are considered in literature, including classifications and index developments. For more information on this topic, the reader can  refer to Kappes et al. (2012).

The analysis uses both absolute and relative aggregations of asset values to identify significant hotspots for population and residential built-up exposure. Choosing absolute or relative spatial data aggregation type has a substantial impact on the results and can therefore lead to distinct information for the decision maker. Addressing both, we leave open the choice for the decision maker to apply  a more utilitarist (Mill, 2007) approach favorising efficiency of measures based on the absolute values or the concept of justice (Rawls, 1971) with prioritisation based on relative aggregations.





The Gi*(d) statistic is used for local spatial autocorrelation analysis using the python-based Exploratory Spatial Data Analysis (ESDA)[6] package. The method describes the spatial autocorrelation as $Z$-score (standard deviations), $p$-value (probability), and confidence level (significance) for each feature (each LAU region). Very high (positive) or very low (negative) $Z$-scores, associated with very small $p$-values (e.g. values of $p < 0.1$), describe spatial clusters as cold spots and respectively hotspots with high significance level.

*Conceptualization of Spatial Relationship.* A known characteristic is that the statistics we are interested in (high $Z$-scores, low $p$-values) are placed in the tails of the distribution and therefore are susceptible to noise and spatial outliers. Moreover, the skewness of a distribution can bias the statistics (Cousineau D.,2010). These are important to consider as the resulting distribution areas of the single hazard clusters needs to be homogeneous in order to be significantly combined in a multi-hazard spatial cluster through meta-analysis (Hak et al., 2016).

To ensure reliable results, the study addresses noise and outliers through a spatial weights matrix. This matrix defines neighboring regions and we use the k-Nearest Neighbour (Fix and Hodges, 1951, Cover and Hart 1967) algorithm which is based on the proximity ($k$) information in order to represent the spatial relationship between regions (LAUs). We have selected this method over contiguity based weights, since the k-nearest neighbour weights displays no ''island'' problem (isolated polygons that do not share any boundaries with other polygons), and every region has at least one neighbour. More information on the factors which affect clustering performance can be found in Zhao, M., et al 2016, on the merits of a weighted matrix.

The study also considers the optimization of spatial autocorrelation/clustering across single hazard exposures by selecting the optimal neighborhood size (k) in the k-Nearest Neighbor (k-NN) algorithm (we present it in Supplementary material section 2).

**2.3 Meta-Analysis: Identifying Regions with Multi-Hazard Potential**

The study adopts a meta-analysis approach to identify regions with multi-hazard potential. This involves combining probabilities ($Z$-scores and $p$-values) from independent hotspots. From the hotspot analysis of different hazards exposure, the same region can show statistically significant positive clustering (hotspot), statistically significant negative clustering (coldspot) as well as statistically non-significant clustering. By the combined outcome of these individual tests that sometimes differ and contradict each other, we measure the multi-hazard potential at regional level. Meta-analysis serves as a viable solution for addressing the challenge of seemingly conflicting evidence in research (Hak et al., 2016; Borenstein et al., 2009). Notably, it serves as a potent tool for conducting robust significance tests (Hak et al., 2016). Consequently, meta-analysis also proves instrumental in resolving the issue of "insignificant results." In the context of our study, meta-analysis serves as a mechanism for synthetizing findings from various clustering analyses. Furthermore, by elucidating the statistical

---

[6] https://pysal.org/esda/




significance of the common estimation, it furnishes an objective "statistical proof" of the potential for multi-hazard clustering in our particular case.

Many *p*-values or *Z*-scores combining methods are used in meta-analysis to aggregate summary statistics. Most used methods are the following:

    i.     Fisher method (Fisher, 1932) based on *p*-value to test the significance of the aggregations;

    ii.    Lancaster's method (Lancaster, 1961) is a generalization of Fisher's test by assigning different weights;

    iii.   Stouffer's method (Souffer, 1949) based on Z-transform test,

    iv.   Lipták's method (Lipták, 1958) which is Stouffer's method with weights, known as weighted Z-test;

    v.    the binomial test (Wilkinson, 1951) which counts the number of *p*-values that are below a threshold α

    vi.   the truncated P-value methods (Zaykin et al., 2002) which adds up *p*-values that fall below a threshold α

For a good overview and comparison of these methods please refer to Whitlock, 2005; Zaykin, 2011; Chen, 2011. Meta-analysis have a widespread use due to their applicability, primarily in psychology, biology and medicine (McFarland, V, L., 2015). Within the field of disaster risk management, meta-analysis have been used mainly to assign the macroeconomy of disasters (van Bergeijk et. Al., 2015).

We chose to use the Stouffer's method (Z-transform test), without weighting, applied on the two-tailed distribution of the single clusters:

$$Z_s = \frac{\Sigma_{i=1}^{k} Z_i}{\sqrt{k}}$$

The sum of *Z*-score*s (Zi)*, divided by the square root of the number of tests, *k*, provides a test of the cumulative evidence on the common null hypothesis (Whitlock, 2005).

Generally the Z-transform test converts the one-tailed *p*-values, from each of *k* independent tests into standard normal deviates *Zi*. A common approach in meta-analysis is to sum the *Z*-scores across studies, weighting them appropriately using the sample sizes. On considering two-tailed method please see Whitlock (2005); Yoon et al., (2021) and on advantages and disadvantages of using the unweighted version of this method please see Becker, B.J. 1994.The z-transform test was performed in python using the *scipy.stats.*[7]

### 3. Results

We identify the regions (LAUs) in Europe at risk to multi-hazards by combining the *Z*-scores and *p*-values across the hotspots of single hazard exposure (i.e. population and built-up) computed on absolute and relative (%) aggregations. In fig.

---

[7] https://docs.scipy.org/doc/scipy/tutorial/stats.html





3, we map these regions and further we consider for a statistical overview the regions with more than 1 hazard exposure (Hz > 1) and confidence level set at 90% (*p*-value < 0.10 and positive *Z*-score > 0).


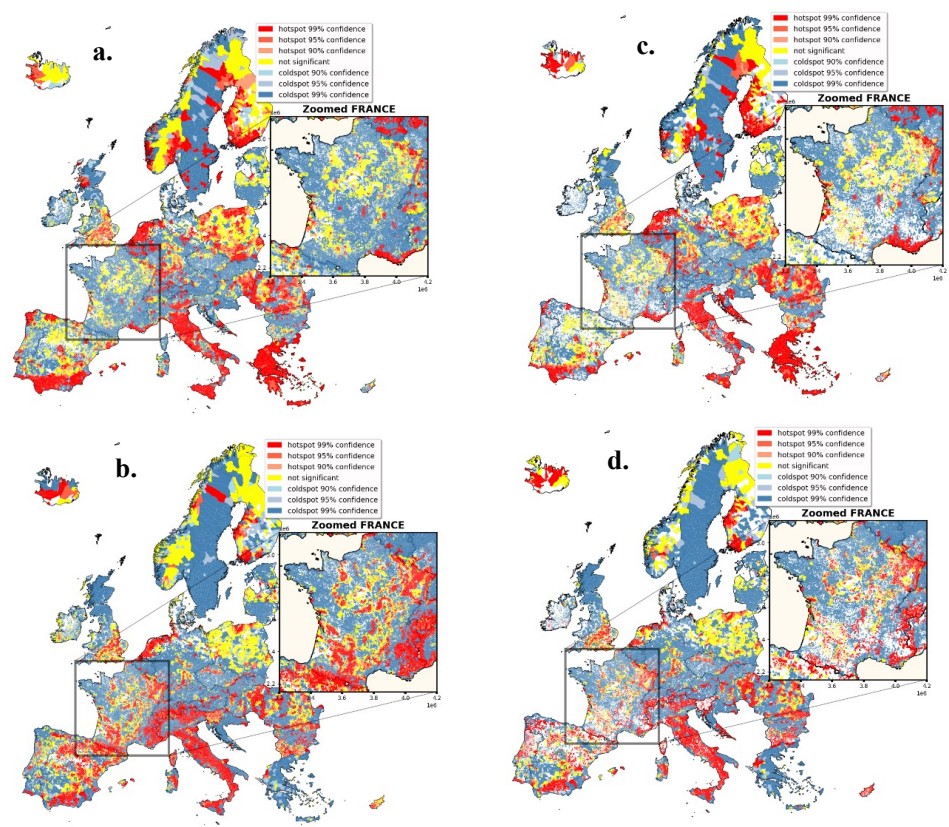

**Figure 3. Regions (LAUs) at risk to multi-hazards identified by the meta-data analysis performed on a). absolute population exposure , b.) relative (%) population exposure, c.) absolute residential built-up exposure and d.) relative (%) residential built-up exposure**


The identification of these regions yielded disparate outcomes contingent upon the specific exposure types scrutinized within our analysis, namely, population density or residential built-up areas. Moreover, the choice of aggregation method, whether relative (expressed as a percentage) or absolute (in terms of the number of individuals or square kilometres of residential built-up areas at risk), introduced variations in both the quantity and spatial arrangement of regions identified as susceptible



to multi-hazard events. A higher number of regions at the European level were identified as susceptible to multi-hazard risks
when considering population-based criteria, as opposed to residential built-up exposures (see Figure 4).Furthermore, there is
a significant difference between the amount of regions being at risk to multi-hazards identified on absolute (12% for
population and 10.6% for residential built-up) compared with the relative aggregation (21% - population and 13.6% -
residential built-up).


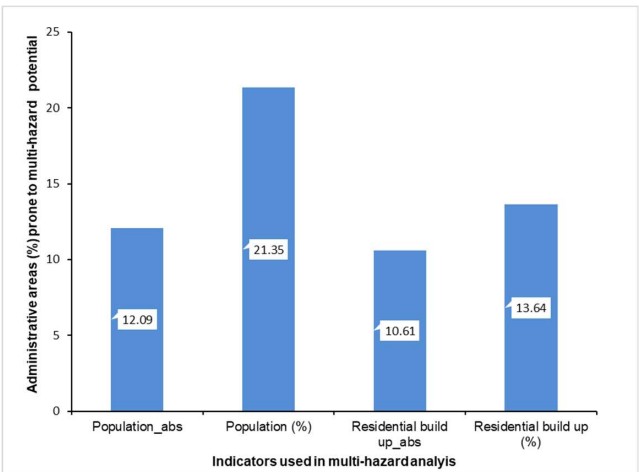

**Figure 4.  Local administrative units (as % from the total in Europe) identified as being prone to multi-hazard risk based on
different indicators  (population and residential built-up) and aggregation types (relative and absolute)**

In order to simplify the interpretation of the results and clearly present the potential of the methodology used, we further
focus only on the regions at risk to multi-hazards identified by the relative (%) population.

We present a statistical overview of these regions identified as being at risk to multi-hazards, looking at their spatial
distribution and their population at risk considering (see 3.1):

i.    various level of economic development (high, middle high, middle low and low income regions - LAUs)

ii.   urbanisation level: rural and urban (according to URAU audit 2018 definitions across European LAUs)

iii.  identifying metropolitan[8]  areas at risk to multi-hazards

iv.   and comparing city centres (city cores – C ) and Functional urban areas (FUA) levels in a metropolitan area

---

[8] The metropolitan areas' according to URAU 2018 definitions and represented here as composed by: core city, Functional Urban Area, Grater city and
Trans-national Functional Urban Area  (codes: C, F, K, T)





### 3.1. Regions (LAUs) with significant multi-hazard potential

Based on population exposure we found 26 058 administrative regions, LAUs (fig. 5) prone to multi-hazard risk in Europe with high significance level (regions with > 90 % confidence interval and number of hazards >1). Most of these regions (20 912) are described statistically as hotspots with highest confidence, 99%, and in only 6 regions in Europe all of the hazard considered for this analysis are present (5 in Italy and 1 in Croatia) (fig. 5 c). These are mountainous and coastal regions..
Regions prone to multiple hazards represents 21.4% of the local administrative units of Europe and around 87 mil. people

(18.8 % of Europe population) (fig. 5 c and d). In figure 5 d, we show that almost half of the population is at risk to more than 3 hazards. Most of these regions are found in France 6956 LAUs, Italy 4627 LAUs, Slovenia 3802, Bulgaria 1876, Spain 1779, Germany and Romania (around 1000 LAUs each). Almost a quarter of the population at risk is in Italy (21.4 mil) and together with the Netherlands (10.1 mil), France (9.5 mil), Spain and Germany (7.1 mil each) they total more than 55% of population at risk to multi-hazards (fig. 5 b and 5 d).


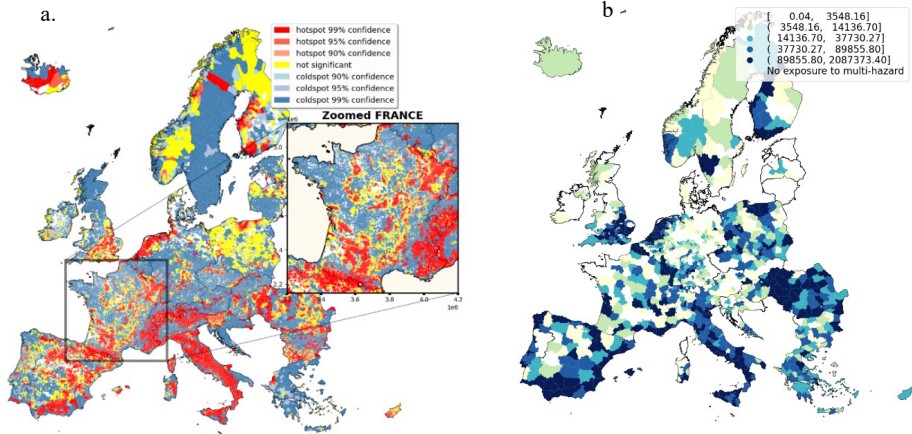



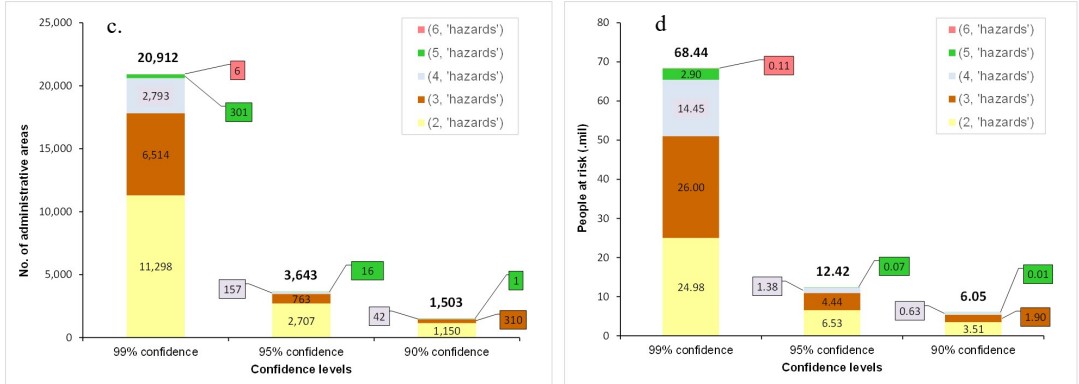

**Figure 5. Regions (LAUs) with population at risk to multi-hazards by significance level (A.) Sum of population at risk to multi-hazards assessed at NUTS3 (only hotspots regions with > 90 % confidence interval); (B.) Number of administrative areas at risk to multi-hazards by confidence interval and number of hazards; (D.) Population at risk to multi-hazards by confidence interval and number of hazards**

*(i.)* In fig. 6 we present the results per income group and degree of urbanisation at European level (fig. 6, a., c.) and by countries (fig. 6 b, d) .

From fig. 6 a., about 36% (9496) of the administrative regions (LAUs) identified as having population at risk to multi-hazards are low income regions and together with the low middle income they sum up to 67%. High income regions represent 10% of the LAUs and high middle income regions 23%. However, the groups of high and high middle income administrative regions total around 50% (43.4 mil) of the population at risk to multi-hazards (fig. 6. c.).

In fig. 6, b., based on income group and degree of urbanisation, we present the top countries with administrative regions (LAUs) identified as being at risk to multi-hazards.

Based on the income groups, most of the high income administrative regions at risk to multi-hazards are in Switzerland (30,9 %), Italy (19.1%) France (16.7%) and Austria, Germany, the Netherlands ( each >5% ) while the low income administrative regions are mostly found in the southern and eastern Europe in Slovenia (31.6%), Bulgaria (19.8%), Romania (10.4%), Hungary (8.9%) and in Italy and Portugal (each > 5%).

In fig. 6, d., most of the low income population at risk to multi-hazards are concentrated in Romania (23%), Italy, Hungary, Poland and Bulgaria (each > 10%) while the high income population at risk to multi-hazards is found in the Netherlands (33%), Germany, Italy and Austria (each >10%).

*(ii.)* Also, from fig. 6, a., the number of administrative areas (LAUs) that are characterised as urban area (based on URAU 2018 definition and on correspondence with LAUs) is much smaller than the number of rural administrative areas (respectively 26.3% or 6585 versus 73.7% or 19200).

Nevertheless, the urban population at risk to multi-hazards total 54% (46.8 mil) compared with the rural administrative areas 46% (40.1 mil) (fig. 6. c.).




Based on the urbanisation degree, 15 countries in Europe have a higher share of population at risk to multi-hazards in rural areas compared to urban areas: Sweden, Norway (100%) or Croatia, Cyprus, Portugal, Slovakia (between 70%-90%) and Hungary, Spain, Belgium, Slovenia, Romania, Switzerland, (between 50%-70%). In the rest of the countries like the Netherlands, Austria (> 80%), Poland, Germany, Greece, (60%-80%), Ireland, United Kingdom, France, Denmark, Czech Republic, Bulgaria (50%-60%) the share of population at risk to multi-hazards in urban areas is higher compared to rural area.

This indicates that people are more at risk to multi-hazards if they live in regions with higher GDP and more densely populated (high and high middle income and urban areas, these are 12% of the administrative regions in Europe) compared with people living in regions with lower GDP and less populated (low and middle low income and rural areas, these are 54% of the administrative regions in Europe). Also considering the degree of urbanisation only, people are more at risk to multi-hazards either if they live in high income urban areas (compared with low income urban areas) or low income rural areas (compared with high income rural areas) (fig. 6. c.).

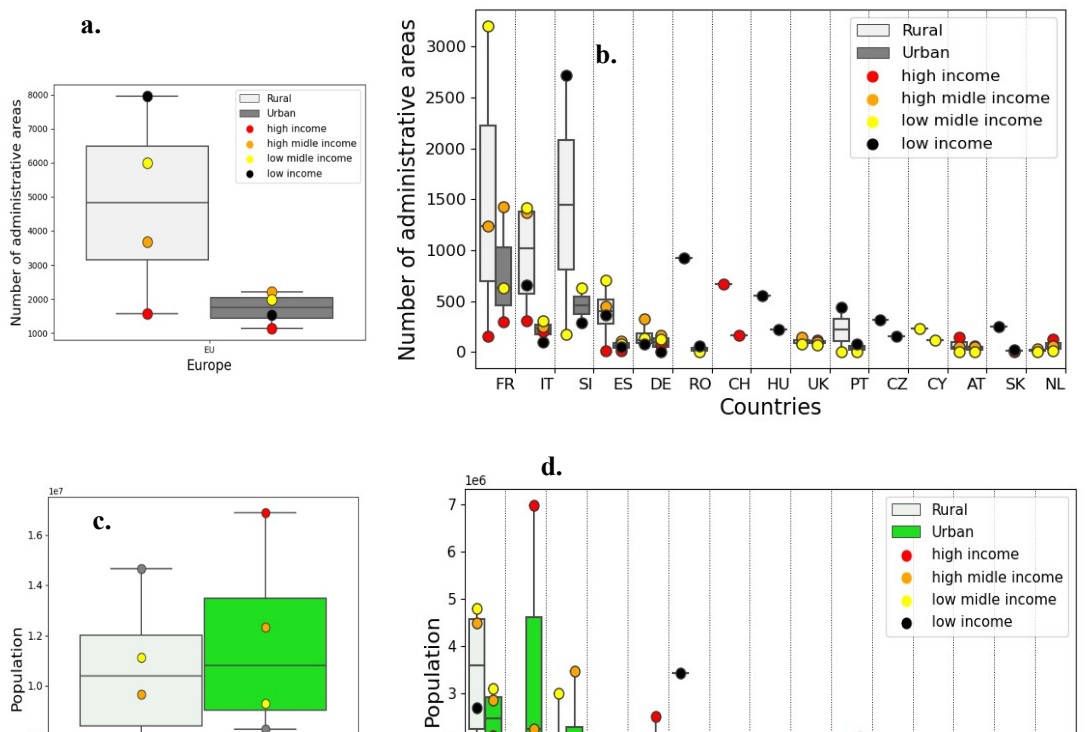





**Figure 6. Number of administrative areas  (LAUs) with population at risk to multi-hazards by income level and urbanization level (A – Europe wide, B -  presents the 15 highest ranked countries)**

From fig. 7 exploring the differences between various income classes we find that as countries and their regions get richer
they get more exposure to multi-hazard risk. After they reach a higher level of income (in the middle income category), the
population at risk from multi-hazard decreases towards the high income. This can suggest that low income countries have the
major part of the population at risk in the rural areas compared to the high income countries where most of the population at
risk is in densely populated urban area and only a quarter from the population at risk  (25% ) live in the rural area. The peak
in the countries with regions in the middle income category could suggest a balance between the high number of urban areas
(the largest across various income classes) and the rural areas with high densities in population.

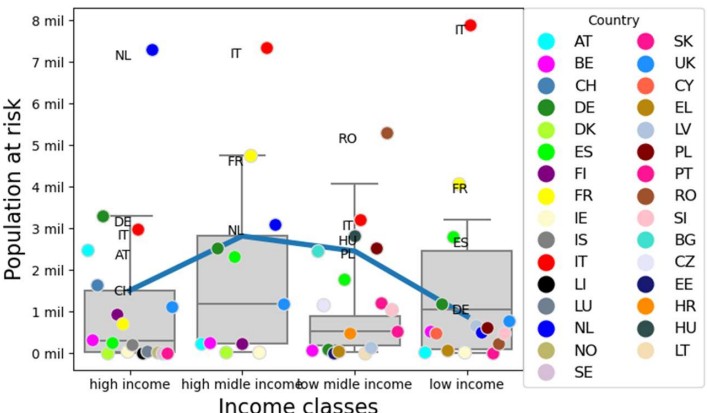

**Figure 7. Population at risk per income level. The markers represent countries' population at risk of multi-hazard split by income**
330                                     **level. The blue line links the 75th quantile of the income classes**

*(iii* Using the Urban Audit 2018 definition and based on correspondence with LAUs we have identified 46% of the
urban/metropolitan areas in Europe (442 of a total of 952) have population at risk to multi-hazards. These urban areas,
totalling 46.8 mil people, are mostly high and middle high income (62.4%). The high income urban areas are mostly found in
the Nederland (28), UK (23), Germany (20), France (9) and Italy (9) while the low income (110 at European level) are found
in Romania (17), Poland (15), Hungary (13), Czech Republic (11) and Bulgaria (16) and others (in Supplementary fig.S23
and table S6)

*(iv.)* In fig. 8 a). and b). we further explore the distribution of population at risk to multi-hazards within the urban areas
comparing the categories: cities (or city cores/centers - C) and larger urbanized zones (commuting zone/Functional Urban -
F). We show that from this local perspective, the population at risk to multi-hazards is governed either by urban population





densities or the expansion of urban land. The commuting zones (FUA) are more at risk to multi-hazards compared with the city centers: 58% (in 257 out of 442) compared with the city centers. However, 57% of the population at risk form the metropolitan areas in Europe live in City Centers. This would suggest that the more population density in the city centers, the more at risk is the metropolitan area. This positive relationships is depicted in fig. 8 b. but is particularly week in the case of high income metropolitan areas, as shown by the almost flat fitting (red) line, and stronger for the middle high and low

income. This shows that going towards the richer metropolitan area the risk increases due to the expansion of the urban area (into the functional urban areas) and diversely, going towards less rich metropolitan areas, due to the densities increase. This is confirmed, with some exceptions (the Netherlands, Austria, Island), by the high income Nordic and Wester countries metropolitan areas where higher proportion of population at risk is found in the functional urban areas compared with the city centers: Denmark, Luxembourg (100%), Finland, Belgium, Switzerland (between 60%-80%) and Ireland, Italy,

Germany and UK (between 50%-60%). Contrarily, in France, Spain and Portugal, most of population at risk (between 50%-60%) is concentrated in city centres of the middle income metropolitan areas which are also the most populated. For the Eastern European lower income countries, the population at risk to multi-hazards is greater in the city centres compared with the functional urban areas: Latvia, Romania, Poland, (> 70%), Bulgaria, Slovenia, Slovakia, Hungary, Czech Republic (between 60%-70%) (Supplementary fig. S24). However, it is evident that the intended comparison could be better explained

through complex urban processes such as changing patterns of residential-choice behaviour due to socio-economic growth which we do not address in this work.

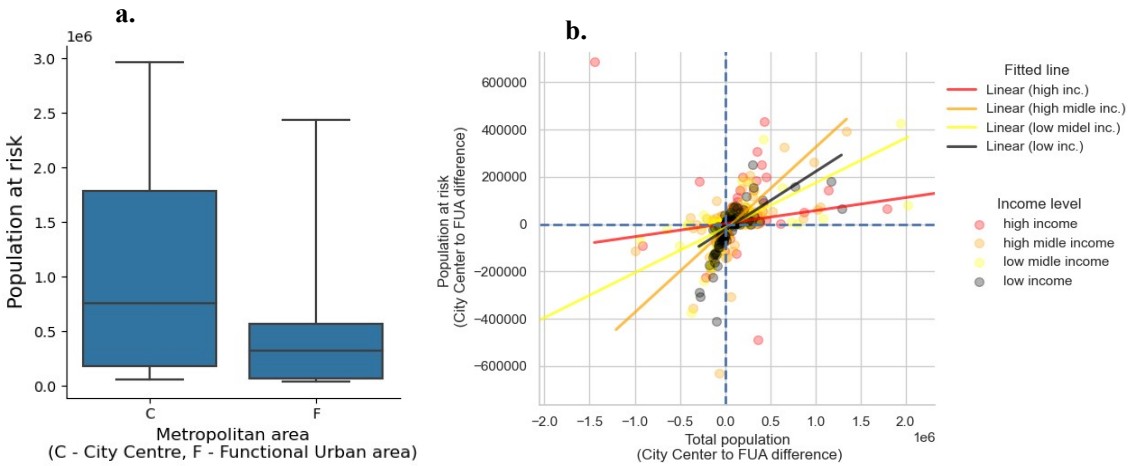

**Figure 8. Population at risk to multi-hazards at the level of Metropolitan area; a). European countries' population at risk within Metropolitan categories: City Centres (C) and Functional urban area (F). The lower and upper whiskers represent, respectively, the lowest 5% and the highest 95% of the calculated population at risk to multi-hazards for each metropolitan category; b). Linear relation between population at risk and total population assessed as difference from FUA of the City Centres.**





## 4.      Validation

The validation proposed is based on  Spearman correlation analysis of the population at risk from multi-hazard with 2 empirical datasets as independent variable: the DRMKC RDH recorded data on fatalities from past events and the count of events with fatalities (for the period 1980-2019), for common hazards: coastal floods, earthquakes, river floods, landslides, subsidence and wildfires. The input data, both the population at risk to multi-hazards and the empirical data are brought to a common geographical scale, the NUTS3 and metric (*Z*-scores and *p*-values of clusters). We use the same methodological

approach explained in this study in order to arrive to single hazard (clusters) hotspots. The single hazard hotspots of empirical data (fatalities and event count) and population at risk to multi-hazards are combined through meta-analysis in order to arrive to a multi-hazard hotspots, of fatalities, event count and of population at risk scaled at NUTS3 level  (fig.9). Finally hot/cold-spots regions of the 2 independent variables (fatalities and event count) are compared with the population at risk from multi-hazard.


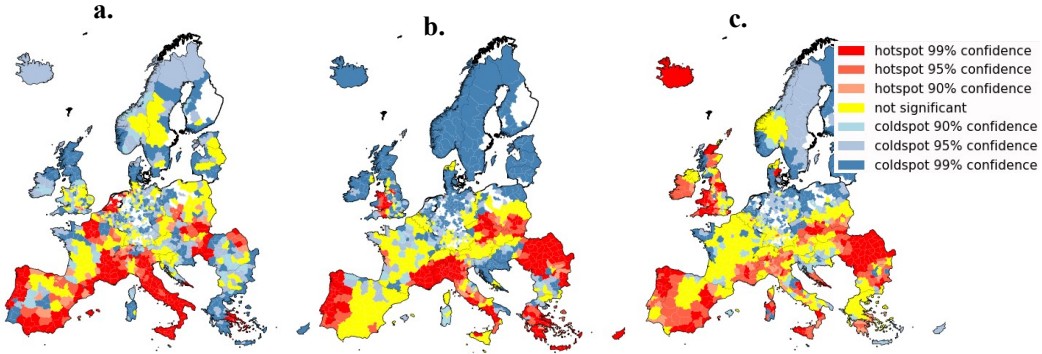

**Figure 9. Identified hot /cold -spots regions (NUTS3) with a.) population at risk to multi hazard; b. ) fatalities from multi- hazard, c. ) number of events with fatalities;  used in  Spearman correlation analysis for the validation purpose**

By using the correlation coefficient analysis we tried to capture the strength of the relation between the two paired datasets, numerically.

We focused on a non-parametric test, the Spearman correlation analysis, because it does not assume that the data is from a specific distribution and is computed on ranks and so depicts monotonic relationships. We choose it as a neutral way of assessing the general central tendency (median) among the pairs of variables at NUTS3. As interpretation, the Spearman

shows the degree by which two variables tend to change in the same direction. Therefore, variables with high correlation increase and decrease simultaneously, while variables with low absolute correlation rarely increase and decrease together.

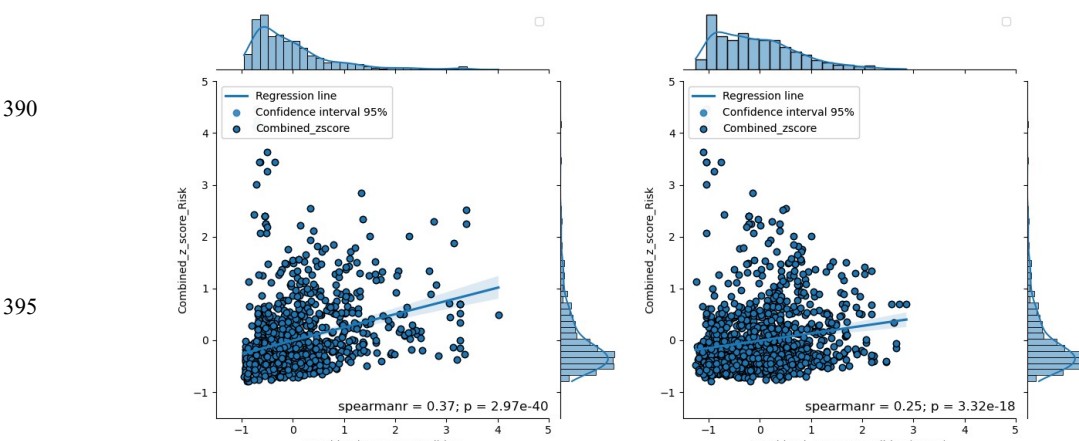



**Figure 10 Spearman correlation between the multi-hazard clusters' size (*Z*-scores) of population at risk with the empirical fatalities from past events (lefts) and events count (right )**

The results presented in fig. 10 refers to the correlation coefficients between paired population at risk with the: a.) amount of fatalities (absolute) and b.) count of events of the empirical data.

We find a rather inconclusive relationship between the multi-hazard risk data and the empirical data, if we consider all regions for all significance levels. The scatterplot suggests a positive correlation between the variables but their increasing monotonic relationship is weak (r=0.37 with fatalities and r=0.25 with the event count) .

However, if we consider only the regions with higher significance (*p<0.01, p<0.05, p<0.10*) we notice a stronger correlation (table 2 and fig. 11). This means that going towards more significant clustering (hot/cold-spots), the independent variables used for the validation tend to follow better the changes in value of the population at risk to multi-hazards.

**Table 2. Spearman correlation coefficient between the empirical data (fatalities and count of past events) and the population at risk from multi-hazard for regions (NUTS3) with different significance levels.**

| Variables | *p_value<0.01* | *p_value<0.05* | *p_value<0.10* | *All regions* |
|---|---|---|---|---|
| Fatalities absolute | 0.59 | 0.51 | 0.46 | 0.37 |
| Count events | 0.30 | 0.40 | 0.35 | 0.25 |

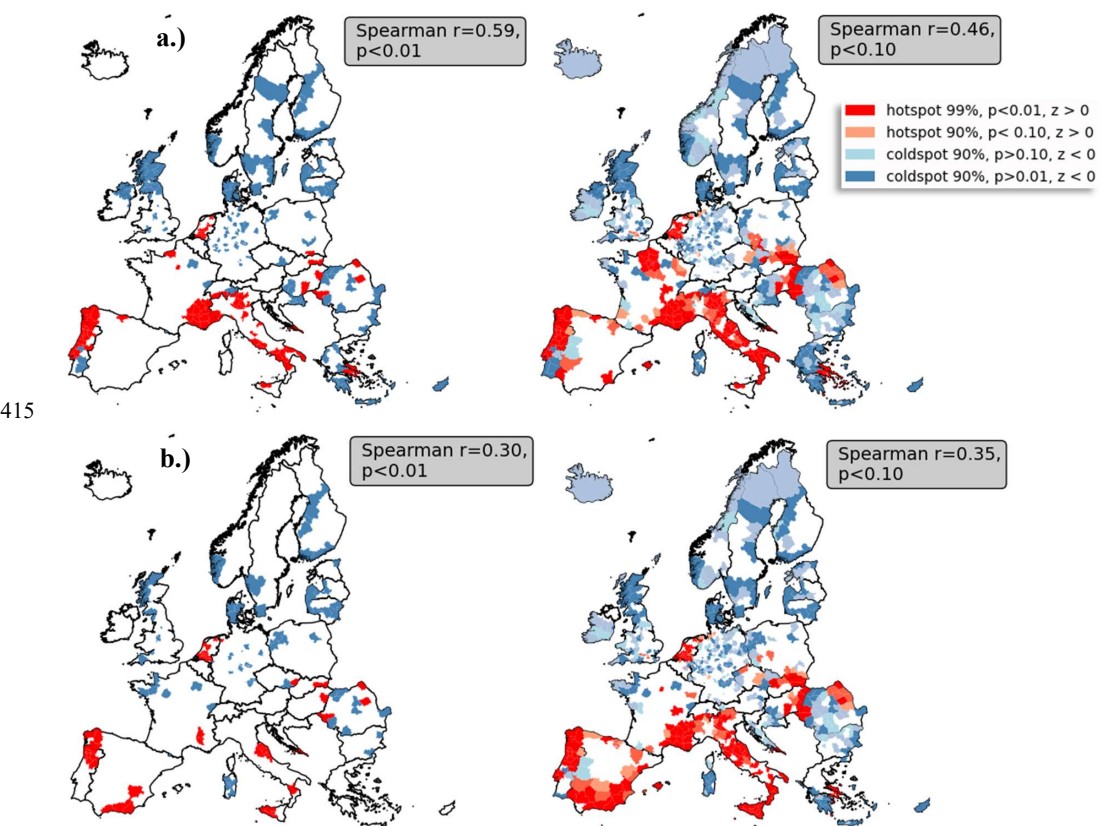


**Figure 11. Regions (NUTS ) at risk to multi-hazards identified with high significance, p<0.01 (left), and p<0.10 (right) as hotspots/cold-spots and their correlation coefficient (Spearman r) with independent variables: a). empirical data – fatalities, b.) empirical data – count of events .**

Therefore, more significant the multi-hazard clustering, stronger is the relationship with the independent variables. The monotonic relationship is strong   r=0.59 with the fatalities as independent variable for the regions with the highest significance p<0.01 while the for the event count the strongest correlation (r=0.40) is reached for the regions with the significance p<0.05. This makes the recorded data on fatalities a better explanatory variable for the clustered population at risk to multi-hazards.




## 5. Discussions

The identification of exposure or risk on the DRMKC RDH platform is done generally from relating an asset to an hazard. There is also the possibility to relate an asset to multiple hazards and have an multi-hazard assessment (of exposure or risk ) on the single asset. This latter situation is the central aspect of the analysis presented in this study which considers the relation of a single asset (e.g. population or the residential built-up) to multiple hazards: landslide, coastal flood, river flood, earthquake, wildfires and subsidence. Starting from this initial setting of the analysis specific characteristics and limitations need to be presented.

First, we show that the proposed methodology allows for the detection of the regions at risk to multi-hazards, differently, as function of the typology of the assets. This is important as it directly reveals relationships between assets types and  threats, valuable for the identification of the disaster risk management pathways in multi-hazard assessment (Ward, P. J., et al 2022).

Furthermore, we argue for an approach that identifies the regions (local administrative units) prone to multi-hazard with high level of significance. We adopt a meta-analysis approach, combining independent tests (the single hazard hotspots),  which seeks to solve the problem of "insignificant results" and provides an objective "statistical proof" of the multi-hazard potential of a region.  We support these results through a validation process which considers empirical data as explanatory variables. We show that more significant is the multi-hazard clustering, stronger is the correlation relationship with the independent variables.

With this study we also show that the proposed methodology allows for detecting changing patterns of the population being at risk from multi-hazard by considering the socio-economic dimension. Our findings are in line with previous studies which present an increase in risk to multi-hazard from low income countries towards higher income countries, and then a decrease as countries' income is the highest (Koks et al 2019). We have also evidenced the highly urbanized regions (urban area) as a space of risk (Hansjurgens et al., 2008) compared with the rural administrative units from multi-hazard occurrence. Furthermore, we show the potential of this methodological approach in detecting the risk to multi-hazard associated with complex socio-economic urban processes. We indicate that high density of population is a good explanatory variables for the increase in risk of the metropolitan areas. However this situation is particularly different in the case of high income metropolitan areas where more at risk to multi-hazards are the population living in the (less densely populated) functional urban area.

We also exemplify the applicability of the multi-hazard methodological approach within the multi-hazard interaction theoretical framework developed by Gill and Malamud (2014). We demonstrate it is scalable down to local and regional level and we present it as one future step in support of DRM decision-making at national and subnational level.

In contrast to other studies, transferability is not limited due to the reliance on case-study-specific data or methodology. The methodological approach described in this study it is already implemented on the DRMKC Risk Data Hub platform (https://drmkc.jrc.ec.europa.eu/risk-data-hub/#/). The methodology utilizes the existing analysis hosted and shared





through the platform and demonstrates its accessibility and replicability. However, differences between the methodological

approach presented and its implementation on the RDH platform exists. They are presented in the section 2.2 of the

Supplementary material.

Whilst we believe that the disaster risk management for multi-hazard assessment is brought forward by the ability

of the proposed approach to identify regions (LAUs) being at risk to multi-hazards with high significance, several

shortcomings are identified.

Most important shortcoming is that the presented case study does not consider the vulnerability for the assessment

of the assets (population and residential built-up) at risk to multi-hazards. The multi-hazard potential of regions is measured

in this study by means of exposure (or assets at risk). Nevertheless, the overall analytical approach is detecting significant

patterns of multi-hazard potential across regions, revealing spatially explicit clusters in a heterogeneous groups of data and

thereby setting the bases for more precise and focused analysis.

Furthermore the clusters are identified at the level of areal dimension (represented by LAUs). The areal dimension

approach excludes a detailed level of study that could more accurately examine the spatial coincidence of multiple hazards at

localized levels. Also, by subdividing the exposure data at the level of areal dimension which are heterogenous in size (see

2.1.1.) will introduce underestimations or overestimations of the clusters especially when the clustering analysis is based on

neighbouring relations defined by distance. However, we identified the optimal $k$ value (dynamic for any relation hazard-

asset) in order to reduce the susceptibility to noise and outliers used in the clustering analysis .

A way of improving the results accuracy and a direction for future research includes the revision of the meta-

analysis (based on the Stouffer's method), used in this analysis. The choice is whether to use the weighted or unweighted

versions of the Z-transform test for Stouffer method when combining the single hazards hotspots into multi-hazard hotspots.

There are arguments in the statistical literature (Whitlock, M.C. 2005) that favour the weighted Z-approach especially when

there is variation in the sample size across studies/clusters (e.g. the number of regions depending on the exposure type) as it

is the case in our study. However the weighted or unweighted version of this test is actually an open question in meta-

analysis (Becker, 1994).

## 6      Conclusions

To our knowledge, this is the first study that uses spatial patterns (clusters/hotspots) and meta-analysis in order to identify

the regions at European level at risk to multi-hazards. The methodology presented in this study provides multi-hazard

enhanced insights, valuable for the identification of the disaster risk management pathways in multi-hazard risk assessments.

The findings point out the socio-economic dimension as determinant factor for the spatial variability and the risk potential of

the local administrative units to multi-hazard. We show that the high density of population is a good explanatory variable for

the identification of the regions at risk to multi-hazards but the economic aspect is the main driver that controls the risk





status at local level: within rural and urban areas and in complex socio-economic urban structure. We also exemplify the applicability of the multi-hazard methodological approach within a multi-hazard interaction theoretical framework and we demonstrate that is scalable to local and regional level and replicable.

By identifying local administrative units with high level of significance as being at risk to multi-hazards we also narrow the 495 uncertainty around the major challenges related with multi-hazard studies: identification of regions prone to multi-hazard, quantification of multi-hazard impact and characterization of the relation between the multiple hazards.

The outcome of this study brings forward an useful methodological input that is made available for use through the Risk Data Hub platform, and is striving to support national authorities on addressing the multi-hazard approach in the National Risk Assessments preparation.







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
