# Peer review of "Spatial identification of regions exposed to multi-hazards at pan European level"

_Natural Hazards and Earth System Sciences, 2023_

## Author Response (AR1)

Author's response

The author appreciates the constructive engagement of reviewers. Each comment offered diverse viewpoints and insights that contributed to a more comprehensive understanding of the topic. The author appreciates the reviewers' feedback which enhanced the quality and depth of the discussion.

**Author's considerations:**

The article describes a methodology for the 'identification of regions at risk to multi-hazard at pan European level'. A statistical approach is used to identify single hazard exposure clusters (hotspots) which further are combined in a multi-hazard exposure clusters (hotspot). A meta-analysis approach is used to identify regions exposed to multi-hazards by combining the probabilities (Z-scores and p-values) of single hazards hotspots. The results are validated using two empirical datasets as independent variables: fatalities from past events and the number of events with fatalities for common hazards (coastal flooding, earthquakes, river flooding, landslides, subsidence, and wildfires).

The author would like to point out that:

1. The analysis takes place in the PREVENTION phase of the DRM cycle, so it assess what could happen, is not based on what has already happen. It is not a prediction it is an assessment of what could happen.
2. The analysis is based on the exposure of assets (population, built-up space) to hazards (coastal flooding, earthquakes, river flooding, landslides, subsidence, and wildfires): exposure/assets at risk = f(Assets, Hazard). Vulnerability is not considered.
3. The analysis provides a methodology to ''cumulate'' or ''combine'' the assets exposed to multiple hazards at Local Administrative Units across EU27+UK+EFTA. It is not a risk assessment and is not assessing any relationship among hazards .
4. The multi-hazard potential of a region is given in this study by the assets deemed at risk/exposed to multiple hazard rather than a quantification of the multi-hazard relations (triggering, cascading etc.) within a region.
5. Our analysis is based on location rather than on areal analysis. We find the exposure or the asset at risk and also the hazard by overlapping the assets (e.g. gridded population, built-up layers) on multiple (gridded) layers describing natural phenomena: river floods, coastal floods, wildfires, subsidence and earthquake (one to many). The overlapping layers provides us on one hand with the exposed assets and on the other hand with the hazard (the peril, the threat), as they define each other, they share the same location  and we show this by intersecting the two information.
    The methodology used is designed to enhance accuracy by directly assessing hazards and exposure in conjunction, reflecting the real-world scenario where these elements interact.
    We propose in this article a methodology which considers that the sole natural phenomenon cannot be defined as hazard/peril/threat and the sole assets are not exposure if they are not assessed against each other.

By clustering the exposure to multiple hazards into spatial patterns and than combining them through meta-analysis we identify the regions at European level at risk to multi-hazards.

**Reviewers comments and reply from the AC.**

COMMENTS: RC 1

- I would consider removing the part of the title behind the colon ': an implemented methodological approach'. It does not add much to the title of the paper, and it would be a stronger, more to-the-point title without it.

  **AC**: change considered

- As a follow-up to the previous comment, I would advise to make the writing in the paper slightly less technical where appropriate (mostly abstract and introduction). As Multi-hazard (risk) is a very interdisciplinary topic, simplifying slightly, without losing the important technical details, will make it more accessible to a broader audience. For example:

  1. Line 15: instead **of *independent tests (the single hazards' exposure hotspots)* only *single hazards' exposure hotspots*** would suffice

  **AC**: correction applied.
  Meta-analysis is defined as a *" statistical technique used to combine the results of multiple independent studies"*, therefore, by using the expression in line 15 we underline that the "the single hazards' exposure hotspots" is the correspondent of the "independent study" from the definition. (The error is that we should use "study" instead of "test".) By removing the "independent study/test" we are concerned that this connection is lost and further on we would not be using meta-analysis anymore but we will 'only' combine 'single hazards' exposure hotspots'. Which is not wrong, but by keeping the relation with the meta-analysis we are also 'holding on' the attributes of the meta-analysis which are : overcoming "non-significant results' and providing objective "statistical proof" Hak et al., 2016. (please see the following comment reply)

  Line 16: ***"insignificant results" and provides an objective "statistical proof"*** comes across as a bit ambiguous, as reader of this abstract, I do not understand what this means.

      1. This is equally unclear in line 59-60 in the introduction. Please clarify.

**AC**: comment considered.

Clarification from (Hack 2016): " In standard practice, meta-analysis is aimed at "solving" the problem that results of studies differ and apparently contradict each other, by generating a "combined" effect. The combined effect might be significantly different from zero (or not). This type of result of a standard practice meta-analysis is considered very valuable because it clearly is a solution to the problem of "contradictory" evidence or, more often, to the problem of "insignificant results". Because meta-analysis functions as a more powerful significance test it generates a more useful and more convincing result than a single study….. The statistical significance of the combined effect might now be interpreted as evidence (or "statistical proof")…."

Therefore, meta-analysis it is valuable because it combines the results of multiple studies, potentially increasing the overall sample size and statistical power. This aggregated data may reveal a significant effect, especially for our study which combines z_scores >0, and p_values <= 0.1.

The sentence could be adjusted as:

'' We adopt a meta-analysis approach, and by combining the hotspots of single hazards' exposure, we address the problem of statistically 'insignificant results'' and we provide an objective "statistical proof" of the multi-hazard potential of a region. ''

2. Line 106-107***: The LAUs are the finest hierarchical classification of statistical regions that together subdivide the European economic territory into regions.*** Would read much nicer and mor accessible as ***The LAUs are the finest hierarchical classification of subdividing the European economic territory into regions in which statistics can be provided at a local level.***

  **AC**: adjustment applied

- Very often throughout the manuscript numbering between brackets is used to. While this works very well on a small scale (Line 50-54 ) to guide the reader, it does not work well in the longer sections such as line 57-77. I understand that you are referring back to the challenges, however It would be better to list the challenges as bullet points (with numbers, Line 50-54) and follow up by clearly stating ***challenge 1 in addressed by ….*** Instead of having ***(1)*** seemingly random in the text.

  **AC**: correction applied

1. Same issue with the overview of the paper (Line 80-85) where line 93-95 seem to be a repetition of 80-83 but with a different numbering system? This is inconsistent.

AC: corrections applied:

"We structure the study as follows, after the *Introduction* we describe the *Data and the methodology,* in *Results* based on identified regions at risk to multi-hazards, we provide a statistical analysis looking at different socioeconomic features. Furthermore, a *Validation* exercise is performed followed by *Discussions* and *Conclusions*. ''

2. Or Line 281, what is this *(i)* referring to? Likewise in Line 331 where *(iii*

**AC**: correction applied.

The statistical overview considers 4 types of areas/administrative regions . This is why we sub-divided the text in 4 parts.

We present a statistical overview of these regions identified as being at risk to multi-hazards, looking at their spatial distribution and their population at risk considering (see 3.1):

i. various level of economic development (high, middle high, middle low and low income regions - LAUs)

ii. urbanisation level: rural and urban (according to URAU audit 2018 definitions across European LAUs)

iii. identifying metropolitan[1] areas at risk to multi-hazards

iv. and comparing city centres (city cores – C ) and Functional urban areas (FUA) levels in a metropolitan area

- In the discussion you have written that one of the large shortcomings of this study is the lack of vulnerability as an input for risk. As a reader this has come to my attention in the result section where a distinction is made between income and urbanization levels, both of which can be seen and used as vulnerability indicators. The statement in line 319-320 **we find that as countries and their regions get richer they get more exposure to multi-hazard risk** is of course true, there are likely assets with higher value present in a hazard-prone area. Yet at the same time these high-income regions may also have more resources to
* * *
[1] The metropolitan areas' according to URAU 2018 definitions and represented here as composed by: core city, Functional Urban Area, Grater city and Trans-national Functional Urban Area  (codes: C, F, K, T)

protect themselves from (multi-) hazards, greatly reducing the This effect may also be visible in Figure 9, take for example the Netherlands, which is a high income hotspot, yet a cold spot when it comes to actual fatalities?  Addressing such vulnerability indicators throughout the results would make the conclusions of the paper more insightful, without having to change the methodology and output.

**AC**: the distinction is made between the LAU regions with income and urbanization levels We have not created an indicator that is included in the 'multi-hazard' analysis. We have only ranked the LAU based on their GDP and urbanization level and we compared the population and residential building exposed to multi-hazards among these regions.

- It is a missed opportunity to reduce all the different hazard types introduced in Table 1 to general multi-hazard risk. While knowing how prone an area is to multi-hazards is indeed a valuable input for national authorities, it is equally important to know which hazard (combinations) they are prone to, for effective DRM plans. Being exposed to droughts and wildfires requires a very different pathway compared to landslides and floods. I am curious to know why this information is not provided? Would it still be possible to showcase this? For example, in a map?

  **AC**: with this analysis it is possible to map the hazard typology. Here  we present the map with all combinations of hazard types present at the level of LAU and depicted by the analysis done on the relative population exposure(the analysis performed on other assets types will presents a different spatial distribution of the hazard types).

[Figure]

*Hazard typology mapped at the level of LAU (CF= Coastal Flood, WUI = wildland urban interface used for wildfire, SUBS = subsidence, LNDSL = landslide, EQ = Earthquake, FL = River flood)*

Even though this analysis is already done, we left it out as we intend to perform a follow up based on this methodology where we will focus on applying the multi-hazard interaction theoretical framework developed by Gill and Malamud (2014). The analysis from our study could be unfolded in a variety of results: considering the number of hazards per region or considering the typologies of hazards . For this study we performed the analysis only considering the high confidence > 90 % interval and number of hazards >1.

We can make available these maps as supplementary material.

**Specific comments**

- Line 41 to 45 you highlight some relevant studies to this research. However, the literature listed has no publications from the last three years, while there have been many advances in this field (e.g. Lee et al. 2024 and Claassen et al. 2023, see references)

  **AC**: this study is a result of a long time process (as in parallel we were implementing the methodology on a web platform). At the moment we start

writing about it we might have overlooked the latest study. But we can add them in the references.

- Line 53 you state the second challenge ***establishment of the impact relation assets – multiple hazards*** What does impact relation assets multiple hazards mean? Can you clarify this or change the wording?

- **AC**:  It means that this methodology allows analysis on the relation single asset paired with multiple hazards (one to many) within a region. This is important as it directly reveals relationships between assets types and  (multiple) threats, it can be quantifiable, and it is valuable for the identification of the disaster risk management pathways in multi-hazard assessment (Ward, P. J., et al 2022).

- You write extensively on the MAUP in line 122-128, yet do not reflect back on it later in the paper. Is this error really not present in your results?

- **AC**: Error could be present as the analysis is geospatial dependent. But by using the smallest area for analysis (as indicated by the references) and by searching the optimal number of neighbors (by identifying the optimal k – parameter) we reduced the error to the smallest  size. "" .. without  completely removing it.''
We didn't make  a validation of the choice as it could be the subject of another study.

- In this paper the distinction between risk and exposure is not clear. For example, in line 131 it states ***The exposure data is built on the relationship hazard (i) - assets (ii).*** And in line 133 you state that significant hotspots of assets at risk are based on two types of exposure aggregations. However, looking at the classic risk question (hazard * exposure * vulnerability) the hazard * assets approach is already leaning more towards risk, but in the paper it is only risk once the exposure is aggregated. Perhaps adding a risk equation will help clarify your approach.

- **AC**: I agree it is to some extent unclear. We express exposure as " assets at risk", defined as assets found in the "footprint" of the hazard. We find the exposure/assets at risk by overlapping the assets (e.g. gridded population) on hazards layers. The overlapped assets 'pixels' are at risk/exposed. Than within an area or region we either sum them up (absolute) or we find their proportion from the total assets of the are/region (% relative aggregation). It gives the sense of the maximum that can be lost in a region.  A formula could be:

  Exposure/assets at risk = $f$(Assets, Hazard)

- Line 154 states that to discriminate the residential typology for built-up and population the CLC 2016 code 1.111 and 2.112 are used. Which land use classes

do these codes refer to? And what purpose does this serve? They don't appear to be referred to later in the paper.

- **AC:** The previous versions GHSL gridded data presents the **amount** of built-up only, but **not the typology**, for the typology we used the CORINE land cover classes. We have presented the CLC 2016 code 1.111 and 2.112. definition in the supplementary the table: Table S2.

- Line 166-170 refers back to the exposure aggregation stated in line 135-136. It would be more concise to only state this once in the methodology to avoid repetition.

  **AC**: the 166-170 has been deleted as it is already presented in the 135-136 as noted by the reviewer

- Line 175-176 cold spots and hotspots are first introduced. Here it would be nice to introduce what each of those means conceptually in terms of risk. A solid definition in the methods also means you don't have to continue to repeat it throughout the results (e.g. line 266).

  **AC**: adjustments made: " In the field of disaster risk reduction and management, identifying both cold spots and hotspots is crucial for allocating resources efficiently. In the present study the hot spots refers to areas or regions with higher susceptibility of risk from multi-hazard while the cold spots can be considered less prone to multi-hazard risks ''

- Figure 8b is difficult to understand. What does do the negative vs. positive values on the x and y axis indicate?

  **AC**: in figure 8 we establish a relationship between the population at risk and the total population, both assessed in relative terms as differences (that is why positive and negative values) from the Functional Urban Area (FUA) of City Centers, categorized by the income.

  Instead of looking at raw population numbers, the analysis is interested in how the population at risk and the total population deviate from what is expected in the context of the Functional Urban Area.

  I added the next part in the description of the figure:

  **"A flatter fitted** line indicates a weaker or less pronounced relationship between the population at risk and the total population. In this case, changes in the total population have a relatively smaller impact on the population at risk. It suggests

that the income category of the region may not strongly influence the risk factors within the population.

**A less flat fitted line**, on the other hand, indicates a stronger relationship between the population at risk and the total population, with changes in the total population having a more significant impact on the population at risk."

It suggests that the income category of the region plays a more prominent role in influencing vulnerability or risk factors within the population. In this scenario, the income level could be a critical factor in determining the extent of risk.

**Technical comments**

- Apostrophes are used in different places throughout the manuscript. For example*, hazards'* in line 15, but *hazard's* in line 40. While both are correct, I would advise to be consistent.

  **AC**: correction applied

- Line 52 says that the methodological approach ***becomes*** the main goal, however, I am guess it ***is*** the main goal. Changing this will make it read better. Additionally in this line I would advise to add 'focused on **addressing** four major challenges '

  **AC**: correction applied

- Figure 1: very minor detail, but I would advise removing the shadow behind the icons and making the icons in the same style (for example all black filled in like the earthquake, landsides, coastal floods and population).

  **AC**: Correction applied

- There appear to be some references missing in the for example intext citations **Liu et al., 2017; Sadegh et al., 2018; Gill et al.,2020** in line 44 are not in the final references list.

  **AC**: correction applied

- Figure 5 appears to have the wrong sub-labeling in the caption. Only a, b, and d are labeled in the caption, and I am not sure if they are labeled correctly. Also, the labels are capitalized in the caption, but not in the figure.

  **AC**: correction applied

- Line 429, **_a hazards_** instead of **_an hazard_**

  **AC**: correction applied

**References**

Claassen, J. N., Ward, P. J., Daniell, J., Koks, E. E., Tiggeloven, T., & de Ruiter, M. C. (2023). A new method to compile global multi-hazard event sets. *Scientific Reports*, *13*(1), 1–14. https://doi.org/10.1038/s41598-023-40400-5

Lee, R., White, C. J., Adnan, M. S. G., Douglas, J., Mahecha, M. D., O'Loughlin, F. E., Patelli, E., Ramos, A. M., Roberts, M. J., Martius, O., Tubaldi, E., van den Hurk, B., Ward, P. J., & Zscheischler, J. (2024). Reclassifying historical disasters: From single to multi-hazards. *Science of the Total Environment*, *912*(October 2023), 169120. https://doi.org/10.1016/j.scitotenv.2023.169120

**COMMENTS : RC 2**

Several questions, comments, and suggestions are raised below concerning terminology, definitions, methodological approach, and pathways to improving clarity and consistency.

1. **Title:** the title of the manuscript does not reflect its content accurately. The methodological approach is essentially developed for multi-hazard exposure assessment. I suggest renaming the paper as 'Multi-hazard exposure assessment at pan-European level' or something similar. Please see the following comments in support of this view.

   **AC:** We have carefully reviewed your suggestions, and after taking into account your supporting comments, we propose a revised title that we believe better encapsulates the focus of our work: "Spatial identification of regions exposed to multi-hazards at pan European level".

We would like to provide further clarification on the concepts used in our study to address any potential doubts:

- Defining  "asset at risk" and exposure:

- In our context, "at risk" is synonymous with exposure, indicating the exposed asset to potential harm, peril , threat - hazard[2].

- Our analysis is based on location rather than on areal analysis. We find the exposure or the asset at risk and also the hazard by overlapping the assets (e.g. gridded population, built-up layers) on multiple (gridded) layers describing natural phenomena: river floods, coastal floods, wildfires, subsidence and earthquake (one to many). The overlapping layers provides us on one hand with the exposed assets and on the other hand with the hazard (the peril, the threat), as they define each other, they share the same location[3] and we show this by intersecting the two information.

- Defining "region at risk"

- The exposed assets or the assets at risk to hazards are further aggregated at regional level across all Local Administrative Units (cca 123 000 LAUs) of Europe. Some LAUs will sum-up exposure to various hazards some will not (NaN). Some regions will be at risk to various hazards some will not.

-Therefore, we determine the presence of multiple hazards in regions and we determine if the regions are at risk or exposed when their assets (in this case gridded population and built-up space) are exposed to these (multiple) hazards.

- 'Multi-hazard Potential impact" and 'Multi-hazard potential of a region'

-Exposed assets or the assets at risk are the maximum that can be lost/impacted. Not all the exposed assets will be impacted/damaged/destroyed, but being
* * *
[2] Gratt, L.B. (1989). "The Definition of Risk and Associated Terminology for Risk Analysis." In Risk Assessment in Setting National Priorities, Advances in Risk Analysis, vol 71.
ISACA. "How to Write Strong Risk Scenarios and Statements." This source emphasizes the importance of scenario building in risk management, which includes understanding the exposure of assets to adverse events 2.

[3] This approach to hazard, as to a spatial distribution of exposed elements is acknowledged in the climate risk and vulnerability assessment frameworks (Schneider et al 2007, Cardona et al 2012 , Birkmann et al 2013) and is in line with the IPCC - influenced perspective of exposure as a spatial concept (e.g. Dunford et al 2015, Preston et al 2007, Veerbeek and Husson 2013).

deemed at risk, as they share spatial location with the hazard, indicates a potential impact on these assets.

However, we acknowledge the need for clarity regarding the distinction between 'multi-hazard potential'  and the 'multi-hazard potential impact' so we are proposing the :

- multi-hazard potential of a region remains as it is, as we assess the presence and threat of multiple hazards on assets (they share the same location), even if we do not assess multi-hazard processes (this is not the scope of the article). The multi-hazard potential of a region is given in this study by the assets deemed at risk/exposed to multi-hazard rather than a quantification of the multi-hazard relations (triggering, cascading etc.) within a region.
- multi-hazard exposure is replacing (as suggested by reviewer) the "multi-hazard potential impact)

2. **Multi-hazard definition**. It would be useful for the reader to understand which 'multi-hazard' definition is adopted in this work to prevent any misunderstandings. There are several variations in the literature, but most studies align to one of the two 'multi-hazard' definitions denoting either (1) the selection of multiple hazards that a region or country faces or (2) the specific context where these hazard events occur simultaneously, cascadingly, or cumulatively over time, and taking into account the potential interrelated effects.  (UNDRR, 2019). This paper addresses multiple hazards without considering their potential interrelated effects (or is a multilayer single-hazard study, as defined by Zschau, 2017), so it subscribes to the first definition. An explicit multi-hazard definition (aligned to the proposed methodological approach) and a few examples of recent multi-hazard events in Europe would enhance the Introduction section.

AC: Though we have presented the 2 definitions in line 43-49 we didn't specify which one of the definitions we adopt. We will add this in line 50:

Our study aligns with the first definition of multi-layer hazards, as we examine the combined exposures of single hazards over a region, recognizing that hazard interrelations can result in an impact distinct from the sum of individual hazards exposures.

- Examples of multi-hazard event reflecting the chosen definition could be (line 53-57):

This is exemplified by events such as the Portugal wildfires and flash floods in October 2017, where these hazards occurred in relatively close succession, both

affecting the same buildings and infrastructure or the floods and the dam failure in summer of 2002 in Czech Republic when floods had caused significant damage to buildings, infrastructure, and agricultural land and subsequent dam failure added to the devastation, impacting structures that were already dealing with the effects of the floods.

3. **Multi-hazard potential as a function of exposure.** The potential or propensity of a region to be affected by (natural) multi-hazard processes is independent of exposure (generally defined as the totality of assets in an area or elements at risk). It is related to specific environmental conditions and factors that increase the likelihood of several hazards occurring in a given time and space. In some parts of the text, this distinction is clear (e.g., lines 162 – 3, for the single hazard hotspot analysis), in others not (e.g., lines 194, 466-7, for the meta-analysis). This introduces significant confusion and also begs the question: why was the exposure analysis done before performing the meta-analysis? Why wasn't the meta-analysis applied to single hazard clusters only? In other words, why are 'single hazards' exposure datasets clustered to identify regions with significant multi-hazard potential (lines 95-6) instead of first performing the meta-analysis on the hazard layers and then combining the results with the exposure layers to indicate multi-hazard exposure? I strongly suggest rethinking this methodological choice and providing an explanation in the text for your decision.

> **AC:** Our methodology is designed to enhance accuracy by directly assessing hazards and exposure in conjunction, reflecting the real-world scenario where these elements interact.
>
> -We propose in this article a methodology which considers that the sole natural phenomenon cannot be defined as hazard/peril/threat and the sole assets are not exposure if they are not assessed against each other. We do this assessment at location level (at lat. long. level by overlapping) and we find hazards and exposure/or assets at risk sharing the same location and even spatial extent. Our analysis is based on location rather than on areal analysis and in the prevention part of the DRM cycle. The impact has not yet occurred, and we are trying to anticipate it by assessing what and where are the potential peril (hazard) and exposed assets located
>
> -We argue that clustering this data and then combining it through meta-analysis is more powerful compared with clustering natural phenomena (which are not hazards) and assets (buildings, population) that are not at risk or exposed (because they are not assessed against each other) and then combining them through the meta-analysis.

-Therefore:

- we determine the potential to multi-hazards by revealing the presence of multiple hazards that are 'threatening' an asset at a common location (the assets is at risk/exposed to multiple hazards because they overlap). We do not assess the multi-hazard processes involved, instead we map the multi-hazard presence by intersecting with assets. The clustering analysis and meta-analysis further creates patterns, grouping together this information and showing where this relation "assets - multiple hazards" forms clusters of threat without assessing various hazards occurring simultaneously or sequentially, cascading at the location.

- also, the multi-hazard processes can be independent of exposure but the hazards cannot be defined independent of the exposure as they define each other, they share the same location.

4. **'Multi-hazard potential impact' and 'multi-hazard potential of a region'**. These two expressions are used interchangeably throughout the manuscript. In the case of the former, the reader would expect some level of information regarding the actual (direct or indirect) consequences or losses associated with multi-hazards. Since this information is not provided (contrary to what is suggested - see lines 54, 496), I recommend using the term 'impact' sparingly and only in the sense indicated above. In most cases, the term 'potential impact' is best replaced with exposure (e.g., lines 12, 53, 54, 68).

**AC**: We understand the concern about the interchangeable use of "multi-hazard potential impact" and "multi-hazard potential of a region."

- In most cases, we will replace "potential impact" with "exposure" to accurately reflect the focus on identifying regions at risk or regions exposed to multi-hazard.

**Specific comments:**

**Introduction:**

1. Line 46: it is more the limited number of sectors rather than 'addressing specific sectors' a limitation in current multi-hazard studies

   **AC:** correction applied

2. Line 53: can you please explain what you mean by 'impact relation assets – multiple hazards'? I suggest removing the reference to impacts (see general comment 4) and replacing it with something similar to 'analyzing exposure to multiple hazards'

   **AC:** The relationship described involves the coupling or matching of assets to hazards, where one element (asset) is related (overlapped) to many elements (single hazards). This relationship offers the advantage of identifying individual assets' exposure distinctly to each hazard. In a given region, an asset may be exposed to a specific hazard but not necessarily to all others.

   **-**Suggested correction applied

3. Line 54: please replace 'quantification of multi-hazard potential impact' with 'quantification of multi-hazard exposure' (see general comment 4)

   **AC:** Suggested correction applied

4. Line 55: is the text referring to a 'common hazard definition' in terms of temporal probability (return period) or the methods by which hazard datasets are derived? Please provide an extra explanation here

   **AC:** A "common hazard definition" demands that all hazards are uniformly characterized, following either deterministic or probabilistic principles. It points out the importance of consistency in the assessment approach, as results derived from deterministic hazard assessments (which involve specific scenarios or predetermined conditions) cannot be combined or compared with those based on probabilistic hazards (which consider the likelihood of various events occurring).

5. References to 'high level of significance', 'independent tests', 'insignificant results', and 'statistical proof' need to be explained as soon as they are introduced. Either postpone their use until the methodological section or provide a short explanation here

   **AC:** Suggested correction considered and the use of the terms was postponed until the methodological section:

   Challenge 1 is addressed in this study with a novel methodology that identifies,at pan-European scale, the regions (Local Administrative Units - LAUs) exposed to multi-hazards with high level of statistical significance (p_value < 0.10). Our approach involves a meta-analysis technique which functions as a powerful significance test (Hak et al., 2016). We combine the hotspots of single hazards'

exposure and we generate a unified result, effectively addressing the challenge presented by divergent and even contradicting independent results. This is the first study that uses spatial patterns (clusters/hotspots) and meta-analysis for this purpose.

6. Gaps analysis in multi-hazard assessment could be better explored in relation to this work

   **AC:** Suggested gaps analysis overviewed:, introduced in line 95:  Research on multi-hazard analysis has underscored several critical gaps that request attention for more effective multi-hazard assessments. These gaps could be listed as : **data quality**, incomplete, outdated, that impacts on accurate multi-hazard assessments (Cutter et al., 2014, Gentile et al., 2022); understanding the **complex interactions** between hazards (Gill et al., 2016; Lee et al., 2024); **temporal dynamics** (Fuchs & Thaler, 2018, De Angelis et al., 2022); addressing **varying vulnerabilities** across hazards (Saaty, 1987; UNISDR, 2004). Additionally, the limited attention given to **uncertainty and sensitivity analyses** in multi-hazard assessments (Haasnoot et al., 2013, Camus et al., 2021); the inadequate **incorporation of climate change** considerations into multi-hazard assessments (IPCC, 2014; Gallina et al., 2016, Ghanbari et al., 2021) and the **communication challenges** conveying multi-hazard risks to stakeholders (Dallo et al., 2020; De Fino et al).

7. It would be good to reference more recent studies (2022 to present) on multi-hazard assessment (multi-layer single hazard)

   **AC:**  We have added : Claassen et al. 2023; Lee et al., 2024

8. Line 60 – 61: please see and reference Classen et al. (2023)

   **AC:** Considered even if as stated before, our study is placed in PREVENTION (way before the impacts happen) and not in RECOVERY on events that already happened. We combine in a multi-hazard analysis, assessments of exposed assets to multiple (perils) hazards. We do not look back at events that already happened and built relationships among hazards.

9. Line 81 (and throughout the paper, including all figure captions): the use of the expression 'regions at risk to multi-hazards' or 'population at risk to multi-hazards' implies that hazard, exposure, and vulnerability are being considered in the methodological analysis. I strongly suggest avoiding the use of 'risk' throughout the paper, as it creates confusion. The meta-analysis results in the identification of regions exposed (not at risk) to multi-hazards. The difference might seem minor but it has methodological implications.

**AC:** As stated previously: In our context, "at risk" is indicating assets, elements, regions deemed at risk, threatened by multi-hazard occurrence. We assessed how many hazards and which hazards are overlapping the assets considered in this study at all locations (lat. long. ) in Europe. E.g.: over one 'pixel' of population we can find 6 hazards pixels overlapping it. So, it is at risk to 6 hazards even if we don't know what kind of multi-hazard processes will be in place.

**Data and methodologies:**

1. Why are (single hazard) exposure clusters (hotspots) instead of multi-hazard clusters assessed (see general comment 3)?

   **AC**: Our methodology is designed to enhance accuracy by directly assessing hazards and exposure in conjunction, reflecting the real-world scenario where these elements interact. It is a methodology which considers that the sole natural phenomenon cannot be defined as hazard/peril/threat and the sole assets are not exposure if they are not assessed against each other. And we do this assessment at location level overlapping these data.

   It is an approach common in Disaster Risk Reduction which is set on identifying the main causal factors of disaster (exposure and vulnerability) that can be managed in relative short time. It is not the hazard the main causal factor of disasters. We present here a methodology that combines exposures (one of these causal factors of disasters) assessed against single hazards (at location level) into a multi-hazard exposure result. The purpose is to identify the regions at European level where the combined and statistical significant outcome shows the multi-hazard peril for those regions. The identifications of these regions is a valuable information that is needed in Disaster Risk Management for prioritization of measures and a better allocation of resources.

   We also acknowledge that considering the relations (cascading, triggering etc.) on hazards and vulnerabilities in the assessment could provide a more accurate outcome.

2. Line 157: Please describe the LU/LC codes directly in the text of the paper

   **AC**: Description introduced in the text …1.111 (continuous urban fabric) and 2.112 (discontinuous urban fabric) **..**

3. Line 159-160: instead of 'regions – LAUs, with assets at risk to single-hazard' a clearer and more accurate description would be 'assets (or elements at risk) exposed to single-hazards'

**AC:** Suggested correction applied.

4. Line 163: instead of 'statistically significant multi-hazard potential of regions' the correct expression would be 'statistically significant multi-hazard exposure of regions'

   **AC:** Suggested correction applied

5. Line 165: Please reference also more up-to-date studies on the combination of multi-layer single hazards using various methodologies

6. Line 175: up until here the paper refers to 'clusters' being synonymous with 'hotspots'; here the paper introduces cold spots as clusters as well; it would be clearer if you used either 'hotspot clusters' (or 'hotspots') and 'cold spot clusters' (or 'cold spots') in the paper rather than just 'clusters' unless you refer to both hot and cold spots at the same time

   **AC:** Clusters and hotspots are similar in that both terms describe localized concentrations or groupings of similar values. However, while "clusters" generally refer to spatial groupings of similar values, "hotspots" / "coldspots" denotes area that exhibits statistically significant clustering, characterized by a *z-score* and a *p-value*. When the *p_vlaue* is small and *z_score* is negative the area is defined as "coldspot", when the *p_vlaue* is small and *z_score* is positive the area is defined as "hotspot". When the p-value is very small (we fixed the *p_values < 0.10* in our study), it means it is very unlikely (small probability) that the observed spatial cluster is the result of random processes (so the spatial pattern denotes a statistically significant clustering).

7. Line 192: Section title refers to 'multi-hazard potential' and not 'multi-hazard potential impact' – although well used in this case, the expression is another evidence of inconsistent use of terminology (see also general comment 4)

   **AC:**

   We acknowledge the need for clarity regarding the distinction between 'multi-hazard potential' and the 'multi-hazard potential impact' so we are proposing the :

   - multi-hazard potential of a region driven by exposure
   - multi-hazard exposure for 'multi-hazard potential impact'

8. Lines 194 – 7: please explain or illustrate how/why the 'independent tests' (i.e., single hazard exposure hotspots) differ and contradict each other and how this issue is solved

   **AC:** From the hot-spot analysis of different single hazards exposure, the same region can show statistically significant positive clustering (hot-spot), statistically significant negative clustering (cold-spot) and also statistically non-significant clustering. We measure the multi-hazard potential exposure of that region by the combining outcome of these 3 individual tests that differ and contradict each other.

   To exemplify, in a region a certain build-up area might be exposed to river flood, coastal flood, landslide, earthquake, wildfire but not to subsidence. The same exposed built-up area can show statistically significant positive clustering (a 'hotspot' with $z\_score > 0$, and $p\_value<0.10$) for river flood, coastal flood, landslide, earthquake but not for wildfire, where it can show a statistically significant negative clustering (cold-spot with $z\_score < 0$, and $p\_value<0.10$). We measure the multi-hazard potential (exposure) of that region by combining the outcome of these individual tests that contradict each other (the same region is described as hotspot and as cold spot depending on how the interested assets is exposed to various hazards).

   (Hack 2016): 'In standard practice, meta-analysis is aimed at "solving" the problem that results when studies differ and apparently contradict each other, by generating a "combined" effect. ..This type of result of a standard practice meta-analysis is considered very valuable because it clearly is a solution to the problem of "contradictory" evidence or, more often, to the problem of "insignificant results. ..... The statistical significance of the combined effect might now be interpreted as evidence (or "statistical proof")...."

**Results:**

1. Lines 246 – 9: please describe here or in the Discussion section what explains the significant difference between the number of regions being exposed to multi-hazards when accounting for absolute versus relative aggregation of elements at risk

   **AC:**

The difference between the number of regions being exposed to multi-hazards when accounting for absolute versus relative aggregation can be attributed to the sensitivity of clustering algorithms to magnitude of the data.

- The clustering algorithm used in our study relies on distance (number of neighbors determined by the optimal *k- parameter* that we have computed in the study) and similarity measures (similar large or small exposure values that are being clustered together).
- Absolute aggregation, based on summed totals, may lead to a clustering process dominated by regions with accentuated absolute variance. This clustering process become susceptible to outliers (very large/small z-scores with high variance) and more concentrated clusters (generally around the LAUs that represents cities and towns) with a smaller number of regions with similar values.
- Relative aggregation, is expressed as percentages of exposure from the total. This allows clustering algorithms to identify patterns based on the proportional contribution of each region, 'smoothing' the dominance of regions with accentuated extreme (large, small) values. As a result, clusters based on relative values may reveal patterns that are more representative across regions, the clusters are smoothed including more regions with similar values. The risk is that the large exposure numbers in densely populated and built-up cities/towns may represent a small proportion compared with their totals and they will not form clusters.

Sentence introduced: The difference in multi-hazard exposure among regions, when considering absolute versus relative aggregation, is influenced by clustering algorithm sensitivity to distance and similarity measures, where absolute aggregation accentuates variance and is susceptible to outliers, while relative aggregation smooths dominance of extreme values, potentially overlooking high-exposure areas within densely populated regions.

2. Line 254: Please remove from Fig. 4 caption the reference to 'multi-hazard risk' and replace it with 'multi-hazards' (see also comment 5i))

   **AC:** suggestion considered

3. Line 308 – 13: These observations should be described in relation to exposure rather than risk, as this is the basis of the results obtained (no conclusions regarding assets being more or less at risk to multi-hazards can be drawn, as vulnerability was not considered; see comment above)

   **AC:** In our context, "at risk" is synonymous with exposure, indicating the exposed asset to potential harm, peril , threat – hazard. We have replaced it with "exposure"

**Validation:**

1. The assumption that the proposed empirical datasets are independent variables doesn't seem to hold: the occurrence or magnitude of one variable (the count of hazard events with fatalities) is likely to influence the other variable (fatalities from past hazard events). For example, if there were more hazard events with fatalities, it is reasonable to expect that the total fatalities from past hazard events would also tend to be higher. Conversely, if there were fewer hazard events with fatalities, the total fatalities would likely be lower. In other words, the probability distribution of one variable does change based on the value of the other variable. Please explain how the two variables are independent or consider using a single variable (e.g., number of fatalities)/a different set of variables

   **AC:** The proposed empirical datasets are independent variables:

   - empirical data on fatalities from past events and (for the period 1980-2019),
   - the count of past events with fatalities (for the period 1980-2019),

   These data are used in a correlation analysis against the population exposed to multi-hazards which is a result of our methodological approach.

**Discussion:**

1. Line 427-8: Exposure is not synonymous with risk – see all other comments on the use of the term 'risk' in this paper

   **AC:** In our context, elements/assets/regions "at risk" is synonymous with exposure, indicating the exposed elements/assets/regions to potential harm, peril , threat, hazard.

2. Line 443: how is the methodology demonstrating the potential to detect 'changing' (i.e., dynamic) patterns of multi-hazard exposure? Please explain or remove the statement

   **AC:** We have compared and explained the distribution of population exposed to multi-hazard in cities (or city cores/centers - C) and larger urbanized zones (commuting zone/Functional Urban - F) within an urban area. We show that from this local perspective, the population exposed to multi-hazards is governed either by population densities or the income. Namely, in high income urban areas the FUA

presents more population exposed to multi-hazards compared to high income city centers (that may show the preference of people to live outside the city centers), while for the lower income the contrary applies, city centers population exposed to multi-hazard is larger compared with the FUA population exposed to multi-hazards. So we have explained this distribution by density and income but it could be other more complex processes that can explain better the distribution of exposed population to multi-hazard in cities and FUA with various income levels:

- environmental sustainability may lead to changing residential choices.
- expansion of public transportation or changes in commuting patterns
- demographic changes, such as an aging population

All these could explain why population chose to concentrate in FUA or City Centers in areas prone to multi-hazards.

This is why the sentence: " the comparison could be better explained through complex urban processes such as changing patterns of residential-choice behavior"

3. Lines 448-9: please explain which complex socio-economic urban processes are considered in the paper; also, please remove the reference to 'detecting risk to multi-hazard' – the paper does not assess risk.

**AC:** Please refer to point 2 above.

4. Lines 453-4 and 491-3: it is not obvious how/where the study illustrates the applicability of the developed approach within the 'multi-hazard interaction theoretical framework' developed by Gill and Malamud, 2014 (and what framework that may be) – please remove statement or provide additional data/information

**AC:** In a previous version of the study we also exemplify the applicability of the methodological approach presented within the multi-hazard interaction theoretical framework developed by Gill and Malamud (2014). Please refer to figures 2, to 11 from the publication for the multi-hazard interaction theoretical framework, which is not included  as it will be the subject of a new study.

The multi-hazard interactions considered in our initial study were:

- primary hazard triggers and increases the probability of secondary hazards, and

- primary hazard increases the spatial overlap and temporal likelihood for triggered secondary hazard

As results we identified:

- Regions (LAU) where primary hazard triggers or increase the probability of the second hazard:  a.) LAUs where River flood (FL) triggers and increases the probability of Landslides (LNDSL) and b.) quantified population (as %) at risk/exposed from this interaction,  b. ) LAUs where wildfire increase the probability of  Landslides (LNDSL) and d.) quantified population (%) at risk/exposed from this interaction  (Maps and plots available);
- Regions (LAU) where primary hazard increases the spatial overlap and temporal likelihood for triggered secondary hazard

5. Lines 454-5: how is the transition from local to regional scale explained? Are LAUs synonymous with regions or are they different? Please ensure consistency in the use of spatial scales

   **AC:** Yes LAUs are the regions, we state this in point 2.1.1:  *For this study the multi-hazard spatial coincidence is assessed at the level of areal dimension, represented by the Local Administrative Units (LAUs).*

6. Lines 459 – 60: There are differences between the methodology described in the paper and the one implemented in the DRMKC RDH which are explained in the SM. However, this is in contradiction with the statements above (lines 457 – 9). It would be clearer to explain that the methodological approach in this paper is an improvement or update of the one existent on the platform and will be made available online soon (including the maps illustrating multi-hazard exposure hotspots which don't seem to be available online at the moment)

   **AC:** We will remove this part from the Article and the SM as the methodology is partially implemented on the platform and currently is work in progress. The table in SM explains the differences and the degree of implementation. At present the clusters are not visible on the platform.

7. Line 465 – 6: Not considering vulnerability in the methodological approach is not a shortcoming but a boundary condition of this paper which does not focus on risk but exposure to multi-hazards; this should be clarified from the title throughout the paper (please see all my other comments on the use of the term 'risk')

**AC:** We have stated that we do not consider the vulnerability. The word 'risk' is used in a context which makes it synonym with the word exposure: therefore "at risk".We will replace this with "exposure" throughout the article.We also present a formula in line 141: exposure/assets at risk = f(Assets, Hazard).

8. Line 470: the scale of analysis (LAUs) is sufficiently large to address questions relating to the spatial (if not temporal) coincidence and interaction (cascading, amplifying, compounding) of different hazards; such studies have been performed at much coarser scales (see Classen et al., 2023, Lee et al., 2024); I suggest stating that this is not within the scope of the current study but could be the object of a future study

   **AC:** Indeed, as presented at point 4. As stated above, Classen et al., 2023, Lee et al., 2024 are reclassifying already happened, historical disasters/events from single to multi-hazards . We do assessment in Prevention, i.e., what could happen.

9. The discussion on limitations is rather short. For example, the authors could reflect more on the effect of combining different hazard scenario definitions (see Table 1) on the accuracy and interpretability of the results

   **AC.** We do not combine hazards, but the effect of their presence (elements becoming at risk or exposed) . And as the element at risk share the same location with multiple perils/hazards we are also aware that there is a multi-hazard presence in regions (exposure and hazards, they define each other as they share the location). The accuracy of the identification of the exposure to multi-hazards is given by the overlapping at the level of pixels of the assets layer with multiple natural phenomena layers.
   The statistical accuracy is given by the significance (rejecting the null hypothesis because we have *p_values* < 0.10 ) of the clusters…which are eventually combined. We also present a validation exercise to observe if our assessment is reflected by the past events impacts or occurrence.

**Conclusions:**

1. Lines 488: please replace 'risk potential' with 'exposure' of LAUs to multi-hazards; similarly for lines 490 and 494

   **AC:** correction applied

2. Lines 495 – 6: please replace 'quantification of multi-hazard impact' with 'multi-hazard exposure'

   **AC:** correction applied

3. Line 496: what do you mean by 'characterization of the relation between multiple hazards'? No relation between the different hazards in this paper is explored; they are treated individually as 'independent tests'; please remove the statement or provide further information

   **AC**: correction applied

4. A more detailed indication of future research work (for example, looking at hazard interrelationships and vulnerability in the context of multi-hazards) would be appreciated.

   **AC**: Future research should aim to address identified limitations by incorporating vulnerability assessments into multi-hazard analyses and the consideration of a multi-hazard interaction framework, for improved accuracy and reliability of multi-hazard hotspots identification.

**Supplementary Material (SM):**

- Please update the Table of Contents (Section 2.2 is missing)

  **AC:** Correction applied

- It would be more useful to show Fig. S22 immediately after Fig. S9, rather than after all elbow method plots

  **AC:** Correction applied

**Technical comments:**

The manuscript is generally well-written, well-referenced, and well-structured. However, many issues with inconsistencies, text composition and formatting, including word choice, sentence readability, punctuation, and spelling are observed both in the paper text and in the SM. I am listing here only those related to the body text of the paper but happy to provide details on specific locations in the SM as well, if necessary. Please make sure the following are addressed:

- Inconsistent use of quotation marks, hyphens, bold text, font themes (see particularly section 2.2 in SM), spaces before punctuation, abbreviations (including 'et al.'), words ('widlfire' – 'forest fire', 'individual' versus 'independent' tests)
- Typos and excessive use of letters for enumerations in text (e.g., (i))
- Use of words (e.g., 'bases' instead of 'basis', 'than'-'then'), superscripts, country abbreviations (e.g., SWE instead of SE, USA instead of Unites States or US)
- Missing explanation of abbreviations (e.g., ESDAC)
- Either the authors use LAUs or regions to refer to the units of analysis
- Missing references (e.g., Gill et al., 2020)

  **AC:** Correction applied

**References:**

Claassen, J. N., Ward, P. J., Daniell, J. et al., (2023) A new method to compile global multi-hazard event sets. Sci Rep 13, 13808. https://doi.org/10.1038/s41598-023-40400-5

Lee, R., White C.J., Sarfaraz Gani Adnan, M., Douglas, J., Mahecha, M.D., O'Loughlin, F.E., Patelli, E., Ramos, A.M, Roberts, M.J., Martius, O., Tubaldi, E., van den Hurk, B., Ward, P.J., Zscheischler, J. (2024) Reclassifying historical disasters: From single to multi-hazards, Science of The Total Environment, 912, 169120, https://doi.org/10.1016/j.scitotenv.2023.169120

UNDRR (2016). Report of the open-ended intergovernmental expert working group on indicators and terminology relating to disaster risk reduction. Terminology available at: https://www.undrr.org/terminology

Zschau. (2017). Where are we with multihazards, multirisks assessment capacities?, in: Science for disaster risk management 2017: knowing better and losing less, edited by: Poljansek, K., Marin Ferrer, M., De Groeve, T., and Clark, I., European Union, Brussels, Belgium. Available at: https://drmkc.jrc.ec.europa.eu/knowledge/science-for-drm/science-for-disaster-risk-management-2017

**Citation**: https://doi.org/10.5194/nhess-2023-220-RC2

Ghanbari, M., Arabi, M., Kao, S.-C., Obeysekera, J., & Sweet, W. (2021). Climate change and changes in compound coastal-riverine flooding hazard along the U.S. coasts. Earth's Future, 9, e2021EF002055. https://doi.org/10.1029/2021EF002055

Fekete, A., Rufat, S., (2023), Should everyone in need be treated equally? A European survey of expert judgment on social vulnerability to floods and pandemics to validate multi-hazard vulnerability factors, International Journal of Disaster Risk Reduction, Volume 85, 103527, ISSN 2212-4209, https://doi.org/10.1016/j.ijdrr.2023.103527.

Gallina, V., Torresan, S., Critto, A., Sperotto, A., Glade, T., Marcomini, A., 2016. A review of multi-risk methodologies for natural hazards: consequences and challenges for a climate change impact assessment. J. Environ. Manage. 168, 123-132, https://doi.org/10.1016/j.jenvman.2015.11.011

Camus, P., Haigh, I.D., Nasr, A.A., Wahl, T., Darby, S.E., Nicholls, R.J., 2021. Regional analysis of multivariate compound coastal flooding potential around Europe and environs: sensitivity analysis and spatial patterns. Nat. Hazard Earth Sys. 21.

Gill, J. C. and Malamud, B. D (2016).: Hazard interactions and interaction networks (cascades) within multi-hazard methodologies, Earth Syst. Dynam., 7, 659–679, https://doi.org/10.5194/esd-7-659-2016, 2016.

De Angeli, S., Malamud, B. D., Rossi, L., Taylor, F. E., Trasforini, E., Rudari, R., A multi-hazard framework for spatial-temporal impact analysis, (2022), International Journal of Disaster Risk Reduction, Volume 73, 102829,
ISSN 2212-4209, https://doi.org/10.1016/j.ijdrr.2022.102829.

Lee, R., White C.J., Sarfaraz Gani Adnan, M., Douglas, J., Mahecha, M.D., O'Loughlin, F.E., Patelli, E., Ramos, A.M, Roberts, M.J., Martius, O., Tubaldi, E., van den Hurk, B., Ward, P.J., Zscheischler, J. (2024) Reclassifying historical disasters: From single to multi-hazards, Science of The Total Environment, 912, 169120, https://doi.org/10.1016/j.scitotenv.2023.169120

Dallo, I., Stauffacher, M., Marti, M., (2020): What defines the success of maps and additional information on a multi-hazard platform?, International Journal of Disaster Risk Reduction, Volume 49, 2020,101761, ISSN 2212-4209, https://doi.org/10.1016/j.ijdrr.2020.101761

De Fino, M., Tavolare, R., Bernardini, G., Quagliarini, E., Fatiguso, F., (2023) *Boosting urban community resilience to multi-hazard scenarios in open spaces: A virtual reality – serious game training prototype*

*for heat wave protection and earthquake response,* Sustainable Cities and Society, Volume 99, 104847, ISSN 2210-6707, https://doi.org/10.1016/j.scs.2023.104847.

Gentile, R., Cremen, G., Galasso, C., Jenkins, L. T., Manandhar, V., Menteşe, E. Y., Guragain, R., McCloskey, J., (2022), *Scoring, selecting, and developing physical impact models for multi-hazard risk assessment,* International Journal of Disaster Risk Reduction, Volume 82, 103365, ISSN 2212-4209, https://doi.org/10.1016/j.ijdrr.2022.103365.

Susan L. Cutter, Kevin D. Ash, Christopher T. Emrich, (2014) *The geographies of community disaster resilience,* Global Environmental Change, Volume 29, 2014, Pages 65-77, ISSN 0959-3780, https://doi.org/10.1016/j.gloenvcha.2014.08.005.

---

## Referee Report (RR1)

Dear Dr. Tiberiu-Eugen Antofie and team

thank you for the revisions you have made to this manuscript. Here are some minor comments that I would still like you to consider to improve the readability of this paper.

**Specific comments**

- Line 51, where it is mentioned that this study looks at multi-layer hazards, is a bit unexpected to the reader, as you mention the study before it has been introduced. It would be more clear to first introducing the study, before highlighting which definition of multi this study considers
- Line 67 is still unclear to the reader. It gets explained in line 81-82, though I wonder if it can not just be stated more clearly in point 2. As 'exposure relationships between assets and multiple hazards.'
- Line 96-104, excellent to have the research gap in the introduction, but this new piece of text feels a bit out of place. As this highlights research gaps, it should go much earlier in the introduction, before highlighting what the study is going to address, before the challenges. Perhaps together with the multi examples in Line 54-58
- Line 268-269 reads a bit confusing, as it says 'compared with city centers' twice to say the same thing? Consider stating the statistics at the start of the sentence. '58% of commuting zones (FUA) are exposed to multi hazards, which is ... higher compared with city centers'
- Line 523-529, it would make more sense to first discuss the outcome of the study and end with future research. The newly added future research appears unconnected to the rest of the text now.

**Technical comments**

- Line 14, should 'develop' be 'developed' as It has been done in the past?
- Line 54-48, make two sentences for readability
- Line 187, the ESDA package, does this need a reference in the footnote? Can this not just be referenced at the end with a normal citation?
- Figure 7 still has risk instead of exposure in it.
- Line 374, still says risk, not exposure.

---

## Author Response (AR2)

**RC1**

Dear Dr. Tiberiu-Eugen Antofie and team

thank you for the revisions you have made to this manuscript. Here are some minor comments that I would still like you to consider to improve the readability of this paper.

**Specific comments**

• Line 51, where it is mentioned that this study looks at multi-layer hazards, is a bit unexpected to the reader, as you mention the study before it has been introduced. It would be more clear to first introducing the study, before highlighting which definition of multi this study considers

**AC:** From lines 40-50, we first provide the context and describe two multi-hazard approaches. In line 51, we highlight our study's alignment with the multi-layer hazards definition. To address the reviewer's concern and improve clarity, we have slightly modified line 51 as follows: *In this context, our study aligns with the first definition of multi-layer hazards, as we examine the combined exposures of single hazards over a region, recognizing that hazard interrelations can result in an impact distinct from the sum of individual hazards exposures*

• Line 67 is still unclear to the reader. It gets explained in line 81-82, though I wonder if it can not just be stated more clearly in point 2. As 'exposure relationships between assets and multiple hazards.'

**AC:** Correction considered at point 2 as suggested

• Line 96-104, excellent to have the research gap in the introduction, but this new piece of text feels a bit out of place. As this highlights research gaps, it should go much earlier in the introduction, before highlighting what the study is going to address, before the challenges. Perhaps together with the multi examples in Line 54-58

**AC:** Correction considered

• Line 268-269 reads a bit confusing, as it says 'compared with city centers' twice to say the same thing? Consider stating the statistics at the start of the sentence. '58% of commuting zones (FUA) are exposed to multi hazards, which is … higher compared with city centers'

**AC:** Correction considered as suggested

*58% of commuting zones (FUA), which is 257 out of 442, are more exposed to multi-hazards compared to city centers*

• Line 523-529, it would make more sense to first discuss the outcome of the study and end with future research. The newly added future research appears unconnected to the rest of the text now.

**AC:** Correction considered as suggested. Ended with future research.

Technical comments

• Line 14, should 'develop' be 'developed' as It has been done in the past?

**AC:** Corrected as '' are developing''

• Line 54-48, make two sentences for readability

**AC:** Corrected, two sentences made:

This is exemplified by events such as the Portugal wildfires and flash floods in October 2017, where these hazards occurred in relatively close succession, both affecting the same buildings and infrastructure or the floods. Similarly, the dam failure in summer of 2002 in Czech Republic when floods had caused significant damage to buildings, infrastructure, and agricultural land and subsequent dam failure added to the devastation, impacting structures that were already dealing with the effects of the floods.

• Line 187, the ESDA package, does this need a reference in the footnote? Can this not

just be referenced at the end with a normal citation?

**AC:** Corrected, introduced as a reference:

Rey, S. J., Anselin, L. (2007). *PySAL: A Python Library of Spatial Analytical Methods*. Review of Regional Studies, 37(1), 5-27.

• Figure 7 still has risk instead of exposure in it.

**AC:** Corrected,

• Line 374, still says risk, not exposure

**AC:** Corrected,

**RC2**

Dear Authors,

Thank you for submitting a revised version of your manuscript which reflects the suggestions and comments proposed. Your considerable effort in providing additional explanations, clarifying misunderstandings, and answering important questions is much appreciated. However, some misunderstandings persist, particularly in relation to concepts like hazard and exposure, which have an important role in your research design and methodological approach. To address these, please can you add/reference the 'exposure' and 'hazard' definitions you use (including the extra explanation that natural events are only termed hazards when they have the potential to harm people or cause property damage, social and economic disruption, etc. and how this specific aspect influenced the methodological approach you proposed).

Below a list of further comments and suggestions:

In the Author's Response:

1. You argue that: 'clustering this data and then combining it through meta-analysis is more powerful compared with clustering natural phenomena (which are not hazards) and assets (buildings, population) that are not at risk or exposed (because they are not assessed against each other) and then combining them through the meta-analysis.'

RC: While I understand the reasoning, the work does not present any evidence of a comparison between the adopted approach and one in which the meta-analysis is performed after assets and hazards layers are combined. Therefore, the argument that the former approach is more powerful cannot be sustained.

**AC:** Thank you for your valuable feedback.

I would like to underline that there are numerous methodologies available for conducting multi-hazard analyses, each with its own strengths and contexts in which it is most effective. The approach we have chosen, which involves first clustering the components of risk (e,g, exposure to single hazards) before combining them through meta-analysis, is one of the well-established methodologies in the field. Please see below some articles reviews underscoring the importance of maintaining the integrity of individual components before their combination.

*Gallina, V., Torresan, S., Critto, A., Sperotto, A., Glade, T., & Marcomini, A. (2016). A review of multi-risk methodologies for natural hazards: Consequences and challenges for a climate change impact assessment. Journal of Environmental Management, 168, 123-132

*de Ruiter, M. C., Ward, P. J., Daniell, J. E., & Aerts, J. C. J. H. (2017). Review article: A comparison of multi-hazard risk frameworks. Natural Hazards and Earth System Sciences, 17(7), 1481-1493.

While it is true that a direct comparison between the two approaches was not performed in this study, the chosen methodology is grounded in established risk analysis practices.

2. In relation to the comment on cold-/hot-spots you provided the following explanation you state: 'while "clusters" generally refer to spatial groupings of similar values, "hotspots" / ''coldspots'' denotes area that exhibits statistically significant clustering, characterized by a z-score and a pvalue. When the p_vlaue is small and z_score is negative the area is defined as ''coldspot', when the p_vlaue is small and z_score is positive the area is defined as ''hotspot''. When the p-value is very small (we fixed the p_values < 0.10 in our study), it means it is very unlikely (small probability) that the observed spatial cluster is the result of random processes (so the spatial pattern denotes a statistically significant clustering).

RC: Please insert this explanation in the text.

**AC:** Inserted, lines180-185

3. In relation to the title of section 2.3, you state: We acknowledge the need for clarity regarding the distinction between 'multihazard potential' and the 'multi-hazard potential impact' so we are proposing the: multi-hazard potential of a region driven by exposure

RC: Another alternative may be: Meta-Analysis: Identifying regions with potential exposure to multi-Hazards

**AC:** Comment considered as suggested

4. In relation to comment 4 in the Discussion section, you state: In a previous version of the study we also exemplify the applicability of the methodological approach presented within the multi-hazard interaction theoretical framework developed by Gill and Malamud (2014). Please refer to

figures 2, to 11 from the publication for the multi-hazard interaction theoretical framework, which is not included as it will be the subject of a new study.

RC: It is not clear to me how Figures 2 - 11 in this manuscript relate with the methodological framework developed by Gill and Malamud (2014). I noted you removed the reference to this study in the manuscript, as it will be the subject of a new study (looking forward to reading it!), so please also remove the reference at lines 518 - 19 ('We also exemplify the applicability of the multi-hazard methodological approach within a multi-hazard interaction theoretical framework')

**AC:** Removed

Please also consider:

- Notation of figures in the text (e.g., line 365 versus 372)

- Missing information - line 376 (Island?)

- Inconsistencies: 'multi-hazard' vs 'multihazard'

- Line 491 - I still argue that the exclusion of assets' vulnerability characteristics is not a shortcoming but a research design choice

**AC:** *Most important shortcoming is that the presented case study does not consider the vulnerability (which was a considered aspect of the research design)for the assessment of the assets (population and residential built-up) exposed to multi-hazards.*

- Reformulating the last sentence in the Conclusions section not as an objective but a potential outcome ('The objective of this study is to support national authorities on addressing the multi-hazard approach in the National Risk Assessments preparation').

   **AC:** All considered as suggested